computational chemistry/physical chemistry

poly(benzimidazole), hydrogen bond, Grotthuss mechanism, DFT method, transition state theory, bifunctional proton transfer

**Author for correspondence:**
Kritsana Sagarik
e-mail: kritsana@sut.ac.th

This article has been edited by the Royal Society of Chemistry, including the commissioning, peer review process and editorial aspects up to the point of acceptance.

# The Grotthuss mechanism for bifunctional proton transfer in poly(benzimidazole)

Jittima Thisuwan[1], Phorntep Promma[2] and Kritsana Sagarik[2]

[1]Division of Science, Faculty of Education, Nakhon Phanom University, Nakhon Phanom 48000, Thailand
[2]School of Chemistry, Institute of Science, Suranaree University of Technology, Nakhon Ratchasima 30000, Thailand

(iD) KS, 0000-0002-3822-4654

Poly(benzimidazole) (PBI) has received considerable attention as an effective high-temperature polymer electrolyte membrane for fuel cells. In this work, the Grotthuss mechanism for bifunctional proton transfer in PBI membranes was studied using density functional theory and transition state theory. This study focused on the reaction paths and kinetics for bifunctional proton transfer scenarios in neutral ($[PBI]_2$), single ($H^+[PBI]_2$) and double-protonated ($H^{2+}[PBI]_2$) dimers. The theoretical results showed that the energy barriers and strength for H-bonds are sensitive to the local dielectric environment. For $[PBI]_2$ with $\varepsilon = 1$, the uphill potential energy curve is attributed to extraordinarily strong ion-pair H-bonds in the transition structure, regarded as a 'dipolar energy trap'. For $\varepsilon = 23$, the ion-pair charges are partially neutralized, leading to a reduction in the electrostatic attraction in the transition structure. The dipolar energy trap appears to prohibit interconversion between the precursor, transition and proton-transferred structures, which rules out the possibility for $[PBI]_2$ to be involved in the Grotthuss mechanism. For $H^+[PBI]_2$ and $H^{2+}[PBI]_2$ with $\varepsilon = 1$, the interconversion involves a low energy barrier, and the increase in the energy barrier for $\varepsilon = 23$ can be attributed to an increase in the strength of the protonated H-bonds in the transition structure: the local dielectric environment enhances the donor–acceptor interaction of the protonated H-bonds. Analysis of the rate constants confirmed that the quantum effect is not negligible for the $N–H^+ \cdots N$ H-bond especially at low temperatures. Agreement between the theoretical and experimental data leads to the conclusion that the concerted bifunctional proton transfer in $H^{2+}[PBI]_2$ in a high local dielectric environment is 'the rate-determining scenario'. Therefore, a low local dielectric environment can be one of the required conditions for effective proton conduction in

acid-doped PBI membranes. These theoretical results provide insights into the Grotthuss mechanism, which can be used as guidelines for understanding the fundamentals of proton transfers in other bifunctional H-bond systems.

# 1. Introduction

Fuel cells have been accepted as environmentally friendly electrochemical devices that can effectively transform chemical energy into applicable electrical energy. However, because the most commonly used polymer electrolyte membrane (PEM) in fuel cells is hydrated Nafion®, the operating temperature has been a serious limitation; water-based fuel cells cannot be operated at and above the boiling point of water. As an aromatic heterocyclic polymer with extraordinary thermal stability and high proton conductivity at relatively high temperatures (400–600 K), poly(benzimidazole) (PBI) has received considerable attention as an effective PEM in fuel cells [1]. Although pristine PBI is not a proton-conducting polymer ($\sigma = 10^{-12}$ S cm$^{-1}$) and cannot be directly used as a PEM [2], acid-doped PBI membranes are excellent proton-conducting polymers [3]. Because the chemistry of acid-doped PBI membranes has been discussed in detail in several review articles [4], only information relevant to the present study is briefly summarized.

Based on the Scatchard method [5], both imide groups in PBI (figure 1) were reported to be preferentially protonated with a considerably high protonation constant compared with sites with lower affinity. The room temperature acid dissociation constants ($K_a$) for these two protonated sites are $5.4 \times 10^{-4}$ and $3.6 \times 10^{-2}$, respectively. It was found that when both imide groups are protonated, proton transfers take place through hydrogen bonds (H-bonds) between the protonated and non-protonated imino nitrogen groups (the N–H$^+$…N H-bond) [6]. Experiments revealed that in anhydrous states in the temperature range between 298 and 433 K, proton conductivities for PBI membranes with acid doping levels of two range from $10^{-9}$ to $10^{-5}$ S cm$^{-1}$ [7].

Acid-doped PBI membranes can be prepared directly from H$_3$PO$_4$ and H$_2$SO$_4$ solutions [8], for which the conductivity depends upon the acid concentration, temperature and duration of immersion [8]. The H$_3$PO$_4$-doped PBI membrane exhibits excellent conductivity. For example, at 403 K, $\sigma$ ranges between $5.0 \times 10^{-3}$ and $2.0 \times 10^{-2}$ S cm$^{-1}$ and is increased to $3.5 \times 10^{-2}$ S cm$^{-1}$ at 463 K [3]. In the high conductivity states, the H$_3$PO$_4$-doped PBI membrane was observed to be very swollen, soft and flexible with a decreased mechanical strength. Analysis of the conductivity in the temperature range of 298 to 403 K revealed that the conductivity of H$_3$PO$_4$- and H$_2$SO$_4$-doped PBI membranes can be explained by using the Arrhenius equation [9], for which the maximum degree of protonation was concluded to be two and one for the acids, respectively, implying that the imide groups in the benzimidazole ring can be fully protonated by H$_3$PO$_4$. It was shown that the high proton conductivity in acid-doped PBI membranes can be explained using the Grotthuss mechanism, not the weak electrolyte theory [9], and $\sigma$ is approximately $10^{-7}$ S cm$^{-1}$ at 303 K [9].

Fontanella *et al.* [10], based on high-pressure conductivity measurements on 600 mol% H$_3$PO$_4$-doped PBI membrane (mediated by the H-bond networks between polymer chains) and 85% H$_3$PO$_4$ solution (mediated by the H-bond networks between H$_3$PO$_4$), suggested that at room temperature, the conductivity decreases with increasing pressure, resulting from an increased viscosity. However, as the temperature is increased, the acid-doped PBI membranes behave like a polymer electrolyte, in which ion transport is enhanced by polymer segmental motions, and, at 348 K, the conductivity increases with increasing pressure. These results indicate that the ion transport mechanisms in a 600 mol% H$_3$PO$_4$-doped PBI membrane and 85% H$_3$PO$_4$ solution are different, especially at high temperature and pressure.

Temperature- and pressure-dependent dielectric spectroscopic experiments also revealed that the PBI molecule is fully protonated after blending with H$_3$PO$_4$ [11]. The results showed that the temperature dependence of the conductivity can be described using the Arrhenius equation, for which the activation energy does not change with H$_3$PO$_4$ concentration. Based on the variation in activation volume with temperature and acid concentration, proton transfer between phosphate and imidazole moieties through H-bonds and the self-diffusion of phosphate are anticipated to be the two most important mechanisms. This rules out the proton transport mechanism through the polymer segmental motion proposed in [10].

In our previous study [12], proton transfer in a unit cell of the imidazole (Im) crystal structure (H$^+$[Im]$_n$, $n = 2$–4) was studied using the density functional theory (DFT) method with the Becke, 3-Parameter, Lee–Yang–Parr (B3LYP) hybrid functional and TZVP basis set. The B3LYP/TZVP results revealed that H$^+$[Im]$_2$ is the smallest, most active Zundel-like complex. Potential energy curves for proton transfer in N–H$^+$…N H-bonds indicated that a single-well potential is favourable in a low

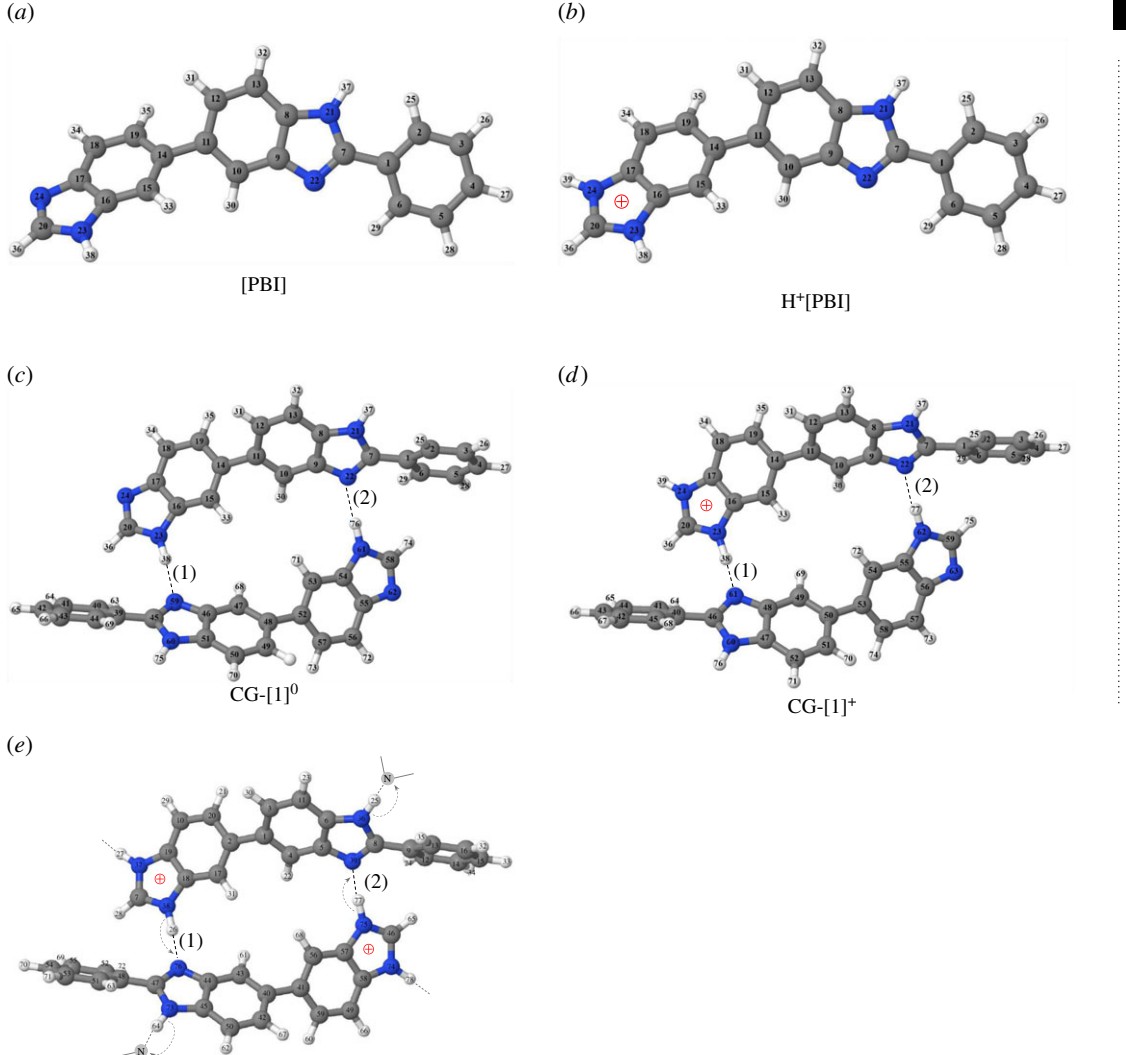

**Figure 1.** Equilibrium structures of the PBI monomers and dimers with atom numbering systems, obtained from B3LYP/DZP and B3LYP/TZP geometry optimizations in the gas phase ($\varepsilon = 1$). The three-character codes are explained in the text. (a) Neutral PBI monomer with C-form ([PBI]). (b) Single-protonated PBI monomer with C-form (H$^+$[PBI]). (c) Neutral PBI dimer ([PBI]$_2$). (d) Single-protonated PBI dimer (H$^+$[PBI]$_2$). (e) Double-protonated PBI dimer (H$^{2+}$[PBI]$_2$) with bifunctional proton transfer scenario.

local dielectric environment ($\varepsilon = 1$), whereas in a high local dielectric environment ($\varepsilon = 23$), a double-well potential dominates, and fluctuation of the local dielectric environment helps promote the Grotthuss mechanism through the Eigen–Zundel–Eigen-like scenario. Born–Oppenheimer molecular dynamics (BOMD) simulations confirmed that the interconversion between single- and double-well potentials (the Eigen–Zundel–Eigen-like scenario) results from the fluctuation of the number of Im molecules in the H-bond chain, and the rate-determining process for proton transfer is a local (short-range) process in which the N–N vibration, oscillatory shuttling motion of the H-bond proton and librational motion of the Im H-bond chain are coherent. Our theoretical studies of the Grotthuss mechanism also revealed that the dynamics of proton transfer can be affected by the motions of aromatic rings, e.g. in phosphonic acid-functionalized polymer (poly(vinyl-phosphonic acid)) [13] and H$_3$PO$_4$-doped imidazole (Im) systems [14]; these motions can be compared with the polymer segmental motion in [10].

According to the above literature review, the following remarks can be made: (i) Experiments studied the temperature dependence of the conductivity of acid-doped PBI membranes using the Arrhenius equation and suggested that proton hopping in acid–acid (e.g. –P–O–H$^+$…O=P–) and acid–PBI H-bond (e.g. N–H$^+$…O) networks dominate especially below 373 K, whereas the information for 'bifunctional proton transfer' in the PBI–PBI (N–H$^+$…N) H-bond chain is restricted; the N-heterocycles in PBI can act as a proton solvent and contribute to proton conduction in the PBI membrane [11]. (ii) There has been controversy as to whether or not proton transfer in a PBI

membrane can be described by the Grotthuss mechanism, for which the Arrhenius equation can be used. (iii) Because the size of the PBI systems is large in view of quantum chemical methods, theoretical studies of proton transfer in a PBI membrane are restricted and the quantum effect has never been previously taken into account; the transportation of small particles such as $H^+$ has been shown in many cases to be governed by the quantum effect (proton tunnelling), for which the rate constants are increased at low temperatures and the tunnelling of protons in the Im H-bond chain along with the $c$ crystallographic direction is anticipated based on solid-state $^{15}N$ NMR spectroscopy [15].

In this study, the Grotthuss mechanism for bifunctional proton transfer in H-bonds between PBI molecules was investigated using the DFT method with the DZP and TZP basis sets. Based on our previous theoretical studies and reported experimental data, neutral ([PBI]$_2$), single ($H^+$[PBI]$_2$) and double-protonated ($H^{2+}$[PBI]$_2$) dimers were chosen as model systems, for which the effect of the size of the basis set and basis set superposition error (BSSE) was initially evaluated. The equilibrium and transition structures and energetics associated with the optimized proton transfer paths in low and high local dielectric environments were computed and analysed in detail; $\varepsilon = 1$ and $\varepsilon = 23$ were included in the model calculations to simulate the lowest and highest effects of the electric field, respectively, for the surrounding PBI molecules on the bifunctional proton transfer process. The kinetics and proton conductivities for the PBI membrane were studied over the temperature range of 200–500 K using transition state theory (TST), and the quantum effect is discussed based on the rate constants obtained with second-order and full Wigner corrections.

# 2. Computational methods

## 2.1. Quantum chemical calculations

Our theoretical studies of the dynamics and mechanisms for proton transfer showed that the DFT method with the B3LYP hybrid functionals was applicable to heterocyclic aromatic compounds [12]. This is due to the ability to approximate H-bond interaction energies with reasonable accuracy and computational resources. It should be noted that the PBI system is a $\pi$ system, in which the $\pi-\pi$ interaction could be important. However, because the dipole moments of the neutral and protonated monomers are relatively high, the dipole–dipole interaction is mainly responsible for the bifunctional H-bond interactions in the coplanar dimers; for example, for the neutral monomer, the dipole moments ($\mu$) in $\varepsilon = 1$ and 23 are 4.1 and 5.9 D, respectively, and those for the single-protonated monomer are 16.3 and 19.8 D, respectively.

In this work, the monomer of the commercially available PBI membrane, [2,2'-($m$-phenylene)-5,5'-bibenzimidazole], shown in figure 1 [16], was used as a model molecule. Because the PBI systems studied in this work are large, the B3LYP method was applied primarily with the DZP basis set (abbreviated B3LYP/DZP), for which the effect of the size of the basis set was systematically investigated using the B3LYP/TZP method and the counterpoise correction of BSSE. The DZP (double zeta polarized) and TZP (triple zeta polarized) basis sets for the H atoms are [2s1p|4s1p] and [3s1p|5s1p] and those for the C and N atoms are [4s2p1d|8s4p1d] and [6s3p1d|10s6p1d], respectively; [A|B] denotes contracted (A) and primitive (B) Gaussian functions.

## 2.2. Equilibrium structures and reaction path optimizations

Our previous studies revealed that for protonated heterocyclic aromatic systems such as Im [12] and $H_3PO_4$-doped Im [14], the elementary processes of proton transfer can be studied reasonably well using presolvation models, in which only important precursor, transition state and proton transferred molecules are taken into account [17]. For the protonated H-bond in the Im system [12], our results also showed that the rate-determining process for proton transfer is characterized by shuttling of the H-bond proton with a vibrational frequency strongly redshifted compared with the neutral H-bond. The theoretical results suggested that for the Im and $H_3PO_4$-doped Im systems, the H-bonds susceptible to proton transfer are strong with extraordinarily short distances, and the smallest, most-active intermediates are protonated dimers.

In this work, because experiments showed that proton transfer in PBI membranes occurs effectively under acidic conditions [2] and theories suggest that the fluctuation of the local dielectric environment can affect the proton transfer potential energy curves, [PBI]$_2$, $H^+$[PBI]$_2$ and $H^{2+}$[PBI]$_2$ in $\varepsilon = 1$ and $\varepsilon = 23$ were chosen as model systems; the thermal energy agitations generally lead to fluctuations of the dipole orientation and local dielectric environment in the vicinity of molecules. The dielectric

constants in the gas phase and bulk were used to simulate (approximate) low and high local dielectric environments, because the most populated value at/in the vicinity of the bifunctional H-bonds and the range over which it fluctuates at a particular temperature are not known.

The equilibrium geometries for the PBI dimers were optimized using the B3LYP/DZP method, and the conductor-like screening model (COSMO) [18] was used to simulate the effect of the local dielectric environment. The strength of the PBI–local dielectric environment interaction was approximated using the solvation energy ($\Delta E^{\text{Solv}}$), computed from the difference between the total energies ($E^{\text{Total}}$) obtained with and without COSMO. In this study, emphasis was placed on $H^{2+}[PBI]_2$ because experiments suggested that both imine groups in the Im rings (figure 1) play important roles in proton transfer in acid-doped PBI membranes, regarded as bifunctional proton transfer [19].

The H-bond interaction energies ($\Delta E^{\text{H-bond}}$) for $\varepsilon = 1$ and $\varepsilon = 23$ were computed from $\Delta E^{\text{H-bond}} = E_{\text{N–H}\ldots\text{N}} - (E_{\text{N–H}} + E_{\text{N}})$, where $E_{\text{N–H}\ldots\text{N}}$ is the total energy of the N–H $\ldots$ N H-bond system and $E_{\text{N–H}}$ and $E_{\text{N}}$ are the total energies of the proton donor and acceptor moieties inside the supermolecule, respectively. To study the effect of BSSE, counterpoise correction [20] was applied, for which $\Delta E^{\text{H-bond/CP}} = E_{\text{N–H}\ldots\text{N}} - (E_{\text{N–H(A)}} + E_{\text{N(D)}})$; (A) and (D) denote the total energies computed with the 'ghost' basis sets (without electrons and nuclei) for the proton acceptor and donor moieties, respectively. The likelihood of proton transfer in H-bonds was anticipated using the asymmetric stretching coordinate ($\Delta d_{\text{DA}}$), which was computed from $\Delta d_{\text{DA}} = |R_{\text{N–H}} - R_{\text{N}\ldots\text{H}}|$; $R_{\text{N–H}}$ and $R_{\text{N}\ldots\text{H}}$ are the N–H and N $\ldots$ H distances in the N–H $\ldots$ N H-bond, respectively. In this study, the H-bond susceptible to proton transfer is characterized by $\Delta d_{\text{DA}}$ close to zero; the H-bond proton located at/or close to the centre of the H-bond is generally governed by a barrierless potential energy [21]. All of the DFT calculations were performed using the TURBOMOLE 7.50 software package [22].

Based on the equilibrium structures and energetics of the PBI dimers, six proton transfer scenarios were hypothesized for $\varepsilon = 1$ and $\varepsilon = 23$. The hypothesized paths were optimized using the nudged elastic band (NEB) method with the Limited-Memory Broyden–Fletcher–Goldfarb–Shanno (L-BFGS) optimizer included in the ChemShell software package [23]. In the reaction path optimizations, 15 images connecting the equilibrium structures of the precursor and product structures were optimized. The relative energies with respect to the precursor ($\Delta E^{\text{Rel}}$), $\Delta E^{\text{H-bond}}$ and $\Delta E^{\text{H-bond/CP}}$ along with the optimized proton transfer paths were plotted against the mass-weighted Cartesians.

## 2.3. Kinetics and proton conductivity calculations

Based on harmonic TST [24,25], the precursor, transition state and product structures obtained from the reaction path optimizations were used in the study of the kinetics of the bifunctional proton transfer process. Because the six proton transfer scenarios hypothesized in this work involve covalent bond dissociation and formation only inside the PBI dimers, unimolecular rate constants were calculated. The classical ($k^{\text{Class}}$) and quantized-vibrational ($k^{\text{Q-vib}}$) rate constants were computed over the temperature range of 200–500 K; these temperatures are lower than the glass transition temperature of PBI (713 K) [26]. $k^{\text{Class}}$ and $k^{\text{Q-vib}}$ were computed to study the possibility to use these semiclassical rate constants in the discussion of proton transfer in H-bond systems. For a one-dimensional energy profile, $k^{\text{Class}}$ is given by [27]

$$k^{\text{Class}}(T) = \frac{k_B T}{h} \frac{Q^{\ddagger}}{Q^R} e^{-\Delta E^{\ddagger}/k_B T}, \qquad (2.1)$$

where $Q^{\ddagger}$ and $Q^R$ are the partition functions of the transition state and precursor equilibrium structures obtained from the NEB method and geometry optimizations, respectively, and $\Delta E^{\ddagger}$ is the energy barrier. $k_B$ and $h$ are the Boltzmann and Planck constants, respectively. For $k^{\text{Q-vib}}$, the energy barrier obtained with the zero-point vibrational energy ($\Delta E^{\ddagger}_{\text{ZPE}}$) is used

$$k^{\text{Q-vib}}(T) = \frac{k_B T}{h} \frac{Q^{\ddagger}_{\text{ZPE}}}{Q^R_{\text{ZPE}}} e^{-\Delta E^{\ddagger}_{\text{ZPE}}/k_B T}. \qquad (2.2)$$

$Q^{\ddagger}_{\text{ZPE}}$ and $Q^R_{\text{ZPE}}$ are the partition functions of the transition state and precursor equilibrium structures obtained with the zero-point vibrational energies, respectively.

To approximate the effect of quantum mechanical tunnelling, the crossover temperature ($T_c$) (the temperature below which the transition state is dominated by quantum mechanical tunnelling) was computed [28,29]

$$T_c = \frac{h\Omega^{\ddagger}}{2\pi k_B}. \qquad (2.3)$$

$\Omega^\ddagger$ is the imaginary frequency of the transition structure. To assess the effect of quantum mechanical tunnelling, the rate constants with quantized vibrations and second-order Wigner correction ($k^{\text{S-Wig}}$) [28,29] were computed based on the assumption that proton tunnelling occurs at the top of the barrier. The Wigner correction to the rate constant was carried out using the Wigner transmission coefficient ($\kappa^{\text{S-Wig}}$)

$$\kappa^{\text{S-Wig}}(T) = 1 + \frac{1}{24}\left(\frac{h\Omega^\ddagger}{k_B T}\right)^2, \tag{2.4}$$

and the Wigner-corrected rate constant is given by

$$k^{\text{S-Wig}}(T) = \kappa^{\text{S-Wig}}(T)k^{\text{Q-vib}}(T). \tag{2.5}$$

$\kappa^{\text{S-Wig}}$ in equation (2.4) is equal to 1 at the classical limit ($h = 0$). In addition, the rate constants with full Wigner correction ($k^{\text{F-Wig}}$) [28,29] were also computed to assess all of the rate constants calculated in this work; although the quantum effect (the proton tunnelling), which generally leads to reduction of the energy barrier, could be studied more accurately using the quantum instanton method, our preliminary reaction path optimizations using this method required extensive computational resources and therefore are not applicable for the present PBI system.

All of the kinetics and thermodynamics calculations were performed using the DL-FIND program [30] included in the ChemShell package [23]. To correlate the theoretical results obtained using the present model systems with the experimental data [6], proton conductivities ($\sigma$) were computed using the Arrhenius equation

$$\sigma(T) = \frac{A}{T}\exp\left(-\frac{\Delta E^\ddagger}{RT}\right), \tag{2.6}$$

where $A$, $T$ and $\Delta E^\ddagger$ are the pre-exponential term, temperature and energy barrier, respectively. $\sigma$ was computed only for the rate-determining processes obeying the Arrhenius equation.

# 3. Results and discussion

The B3LYP/DZP and B3LYP/TZP methods suggested that the equilibrium structures of the [PBI], $H^+$[PBI] and $H^{2+}$[PBI] monomers take on approximately the same planar structure (C-form in figure 1), and the structures of the dimers formed from these monomers are also an approximately planar structure (electronic supplementary material, table S1); the high proton conductivity of the PBI membrane was attributed to the linear PBI chains with a high tendency for coplanar aromatic rings [19]. Therefore, the discussion, henceforth, is focused on these C-form dimers. In this study, the Grotthuss mechanism was hypothesized to occur through the bifunctional proton transfer process in the coplanar H-bond networks, e.g. in $H^{2+}$[PBI]$_2$, as shown in figure 1e.

For ease of discussion, structures of the PBI dimers are labelled with a three-character code, e.g. **CG-[m]**$^n$ and **CC-[m]**$^n$; **CG** = dimer with C-form in $\varepsilon = 1$; **CC** = dimer with C-form in $\varepsilon = 23$; $n$ = total charge (the number of protonated H-bonds). The same or different dimer structures are distinguished by **[m]**. For example, **CG-[1]**$^+$ and **CC-[1]**$^+$ are $H^+$[PBI]$_2$ with the same structure ([1]$^+$) in $\varepsilon = 1$ (**CG**) and $\varepsilon = 23$ (**CC**), respectively. **CG-[1]**$^0$ and **CG-[2]**$^0$ are [PBI]$_2$ in $\varepsilon = 1$ with different structures, [1]$^0$ and [2]$^0$, respectively. In addition, $\ddagger$ refers to the transition structures on the potential energy curves.

## 3.1. The DFT methods and equilibrium structures

The total ($E^{\text{Total}}$) and H-bond interaction energies ($\Delta E^{\text{H-bond}}$ and $\Delta E^{\text{H-bond/CP}}$) obtained from B3LYP/DZP and B3LYP/TZP calculations are listed in the electronic supplementary material, table S1, together with the H-bond distances ($R_{\text{N–N}}$) and asymmetric stretching coordinates ($\Delta d_{\text{DA}}$). Both B3LYP/DZP and B3LYP/TZP geometry optimizations yield the same dimer structures. For $\varepsilon = 1$, the $R_{\text{N–N}}$ distances obtained from the B3LYP/DZP method are slightly shorter with stronger H-bonds (more negative $\Delta E^{\text{H-bond}}$). Although the stability of **CG-[1]**$^{2+}$ is higher (more negative $E^{\text{Total}}$) than **CG-[1]**$^+$ and **CG-[1]**$^0$, the DFT methods suggest that the H-bonds in **CG-[1]**$^+$ are stronger than those in **CG-[1]**$^0$ and **CG-[1]**$^{2+}$. The B3LYP/DZP results give $\Delta E^{\text{H-bond}}$ = −171.2 (−151.4) kJ mol$^{-1}$, −84.8 (−69.4) kJ mol$^{-1}$ and −83.6 (−62.1) kJ mol$^{-1}$, respectively; hereafter, the values obtained from the B3LYP/TZP method are shown in parentheses. The H-bond stability order is in accordance with $R_{\text{N–N}}$ and $\Delta d_{\text{DA}}$ in

the electronic supplementary material, table S1, in which the H-bond **(1)** in **CG-**[1]$^+$ is the shortest, $R_{N-N} = 2.68$ (2.72) and $\Delta d_{DA} = 0.46$ (0.54) Å.

The counterpoise correction does not lead to any change in the H-bond stability order; $\Delta E^{\text{H-bond/CP}} = -151.8$ (−147.1) kJ mol$^{-1}$, −66.9 (−65.3) kJ mol$^{-1}$ and −63.6 (−57.8) kJ mol$^{-1}$, respectively. Comparison of $\Delta E^{\text{H-bond}}$ and $\Delta E^{\text{H-bond/CP}}$ reveals that the discrepancies in the H-bond interaction energies obtained from the B3LYP/DZP and B3LYP/TZP methods are significantly reduced when the counterpoise correction is included; for example, for **CG-**[1]$^+$, without the counterpoise correction, $\Delta\Delta E^{\text{H-bond}} = 19.8$ kJ mol$^{-1}$, and with the counterpoise correction, $\Delta\Delta E^{\text{H-bond/CP}} = 4.7$ kJ mol$^{-1}$.

For $\varepsilon = 23$, B3LYP/DZP and B3LYP/TZP geometry optimizations suggest the same stability order as for $\varepsilon = 1$ with solvation energies of $\Delta E^{\text{Solv}} = -494.3$ (−502.8) kJ mol$^{-1}$, −226.5 (−235.1) kJ mol$^{-1}$ and −106.9 (−113.8) kJ mol$^{-1}$ for **CC-**[1]$^{2+}$, **CC-**[1]$^+$ and **CC-**[1]$^0$, respectively. The values for $\Delta E^{\text{Solv}}$ are correlated with the net positive charges of the H-bond dimers H$^{2+}$[PBI]$_2$, H$^+$[PBI]$_2$ and [PBI]$_2$. The results given in the electronic supplementary material, table S1 reveal that the N–H … N H-bonds are weaker for $\varepsilon = 23$ with different H-bond stability orders compared with $\varepsilon = 1$. The H-bonds in H$^{2+}$[PBI]$_2$ are stronger than those in H$^+$[PBI]$_2$ and [PBI]$_2$; $\Delta E^{\text{H-bond}}$ for **CC-**[1]$^{2+}$, **CC-**[1]$^+$ and **CC-**[1]$^0$ is −67.9 (−42.2) kJ mol$^{-1}$, −62.6 (−39.1) kJ mol$^{-1}$ and 46.2 (−25.3) kJ mol$^{-1}$, respectively. The H-bond stability order follows the trend for $\Delta E^{\text{Solv}}$: the lower the $\Delta E^{\text{Solv}}$ value (more negative), the stronger the H-bonds.

While the counterpoise correction leads to a decrease in the strength of the H-bonds in **CC-**[1]$^+$ and **CC-**[1]$^0$ (as in the case for the uncorrected BSSE), the counterpoise correction increases the strength of the H-bonds in **CC-**[1]$^{2+}$; $\Delta E^{\text{H-bond/CP}} = -107.8$ (−103.1) kJ mol$^{-1}$ compared with $\Delta E^{\text{H-bond}} = -67.9$ (−42.2) kJ mol$^{-1}$. It appears that with the counterpoise correction, the discrepancy between the H-bond interaction energies obtained from the two DFT methods is significantly decreased (as already observed for $\varepsilon = 1$), e.g. for **CC-**[1]$^{2+}$, $\Delta\Delta E^{\text{H-bond/CP}} = 4.7$ kJ mol$^{-1}$. Therefore, one can conclude that $\Delta E^{\text{H-bond/CP}}$ obtained based on the B3LYP/DZP and B3LYP/TZP methods with the counterpoise correction are approximately the same, and it is reasonable to use the B3LYP/DZP results with the counterpoise correction in further discussion.

Because the local dielectric environment affects charges on atoms involved in H-bonds, to study this effect on the strength of H-bonds, the net stabilization energies ($\Delta E^{\text{NSE}}$), defined as the difference between $\Delta E^{\text{H-bond/CP}}$ for $\varepsilon = 23$ and 1, were calculated, and are given in the electronic supplementary material, table S1; H-bonds are stabilized for $\varepsilon = 23$ when $\Delta E^{\text{NSE}}$ is negative. The values for $\Delta E^{\text{NSE}}$ show that the equilibrium structure of H$^{2+}$[PBI]$_2$ is strongly stabilized in a high local dielectric environment, whereas that for H$^+$[PBI]$_2$ and [PBI]$_2$ is moderately and strongly destabilized, respectively; $\Delta E^{\text{NSE}} = -44.2$, 3.8 and 61.1 kJ mol$^{-1}$, respectively.

To complete the evaluation of the DFT methods, the optimized proton transfer paths for bifunctional proton transfer in **CG-**[1]$^0$ are discussed as an example. Because the structure of the potential energy curve governs the kinetics of proton transfer, the relative total energies ($\Delta E^{\text{Rel}}$) obtained based on the B3LYP/DZP and B3LYP/TZP methods are compared in the electronic supplementary material, figure S1. Additionally, the relative H-bond interaction energies with respect to the precursor **CG-**[1]$^0$ ($\Delta E^{\text{Rel,H-bond}}$ and $\Delta E^{\text{Rel,H-bond/CP}}$) are included in the electronic supplementary material, figure S1; $\Delta E^{\text{H-bond}}$ and $\Delta E^{\text{H-bond/CP}}$ for **CG-**[1]$^0$ obtained from the B3LYP/DZP and B3LYP/TZP methods are set to 0 kJ mol$^{-1}$. It appears that the $\Delta E^{\text{Rel}}$ values obtained from both methods are approximately identical and $\Delta E^{\text{Rel,H-bond}}$ and $\Delta E^{\text{Rel,H-bond/CP}}$ are virtually the same. These results confirm the applicability of the B3LYP/DZP method with the counterpoise correction, and the discussions, henceforth, are based only on these methods.

## 3.2. Potential energy curves for bifunctional proton transfer

The transition structures and energetics for the optimized bifunctional proton transfer paths are included in the electronic supplementary material, table S2. The potential energy curve in figure 2$a$ and the energy values in table 1 show that for $\varepsilon = 1$, bifunctional proton transfer in [PBI]$_2$ is non-concerted and not preferential in the forward direction (a strong uphill process). Starting from **CG-**[1]$^0$ ($\Delta E^{\text{H-bond/CP}} = 66.9$ kJ mol$^{-1}$), proton transfer first takes place in the H-bond **(1)**, leading to the ion-pair transition structure **CG-**[2]$^{0,\ddagger}$ ($\Delta E^{\ddagger} = 97.3$ kJ mol$^{-1}$), and the second proton transfer in the H-bond **(2)** ($\Delta E^{\ddagger} = 162.3$ kJ mol$^{-1}$) generates the structure **CG-**[3]$^0$ as the product structure. Structure **CG-**[2]$^{0,\ddagger}$ is characterized by an extraordinarily strong ion-pair H-bond ($\Delta E^{\text{H-bond/CP}} = -556.7$ kJ mol$^{-1}$), and **CG-**[3]$^0$ is represented by double ion-pair H-bonds ($\Delta E^{\text{H-bond/CP}} = -356.7$ kJ mol$^{-1}$). The ion-pairs resulting from the acid–base interaction have been suggested to possess H-bond characters, for which the strength of the association depends on the nature of the cation: the smaller the cation size, the stronger the ion-pair association [31].

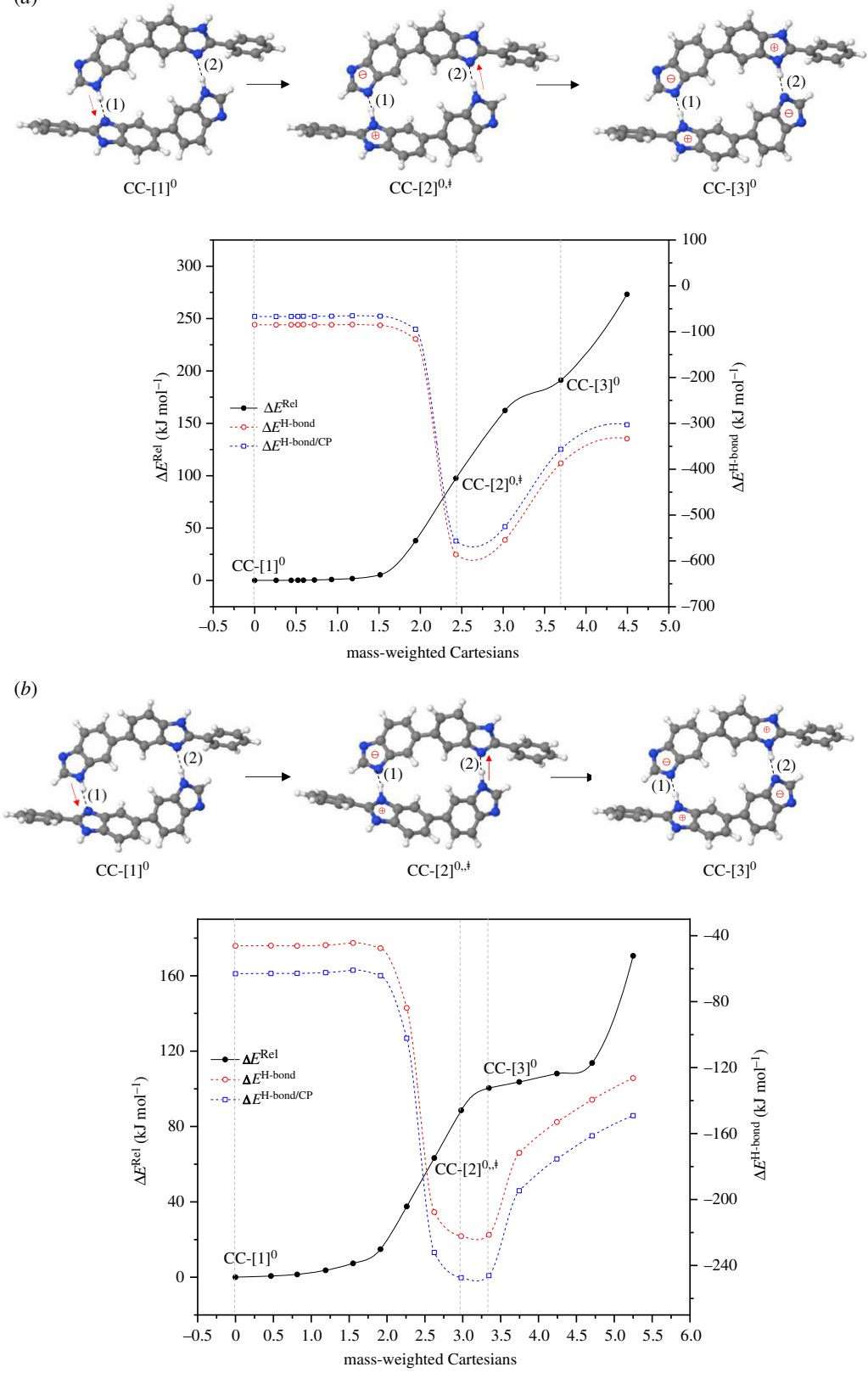

**Figure 2.** Reaction path and H-bond interaction energies for bifunctional proton transfer in the neutral PBI dimer obtained based on the B3LYP/DZP and NEB methods. The three-character codes are explained in the text. (*a*)–(*b*) In the gas phase ($\varepsilon = 1$) and continuum dielectric environment ($\varepsilon = 23$), respectively; $\Delta E^{\text{Rel}}$ = relative total energy; $\Delta E^{\text{H-bond}}$ = H-bond interaction energy; $\Delta E^{\text{H-bond/CP}}$ = H-bond interaction energy obtained with the counterpoise correction of the BSSE; $\Delta E^{\ddagger}$ = energy barrier of proton transfer; x-axis = mass-weighted Cartesians.

**Table 1.** Rate constants ($k$), energy barriers without ($\Delta E^\ddagger$) and with the zero-point vibrational energy corrections ($\Delta E^\ddagger_{ZPE}$), activation enthalpy ($\Delta H^\ddagger$) and Gibbs free energies ($\Delta G^\ddagger$) for bifunctional proton transfer in the PBI dimers. Energies, rate constants and temperatures are in kJ mol$^{-1}$, s$^{-1}$ and K, respectively. $k^{Class}_{f/r}$ = rate constant obtained from classical TST; $k^{Q\text{-vib}}_{f/r}$ = rate constant obtained with quantized vibrations including the zero-point vibrational energy; $k^{S\text{-Wig}}_{f/r}$ = rate constant obtained with quantized vibrations and tunnelling correction through the simplified Wigner correction to the second order; $k^{F\text{-Wig}}_{f/r}$ = full Wigner-corrected rate constant at $T$ above $T_c$. f/r = forward or reverse direction.

| channel | $\Delta E^\ddagger$ | $\Delta E^\ddagger_{ZPE}$ | $\Delta H^\ddagger$ | $T_c$ | $T$ | $k^{Class}_{f/r}$ | $k^{Q\text{-vib}}_{f/r}$ | $k^{S\text{-Wig}}_{f/r}$ | $k^{F\text{-Wig}}_{f/r}$ | $\Delta G^{\ddagger,(Rel)}$ |
|---|---|---|---|---|---|---|---|---|---|---|
| CG-[1]$^+$ → CG-[2]$^{+,\ddagger}$ | 1.8 | −6.2 | −7.0 | 191 | 200 | $5.32 \times 10^{12}$ | $1.07 \times 10^{14}$ | $2.68 \times 10^{14}$ | $2.29 \times 10^{15}$ | (−5.4) |
|  |  |  |  |  | 333 | $8.27 \times 10^{12}$ | $3.49 \times 10^{13}$ | $5.37 \times 10^{13}$ | $6.45 \times 10^{13}$ | (−4.5) |
|  |  |  |  |  | 500 | $1.03 \times 10^{13}$ | $2.20 \times 10^{13}$ | $2.72 \times 10^{13}$ | $2.83 \times 10^{13}$ | (−3.1) |
| CG-[2]$^{+,\ddagger}$ ← CG-[3]$^+$ | 12.8 | 2.2 | 1.2 | 191 | 200 | $7.43 \times 10^{9}$ | $6.34 \times 10^{11}$ | $1.58 \times 10^{12}$ | $1.35 \times 10^{13}$ | 3.1 |
|  |  |  |  |  | 333 | $1.60 \times 10^{11}$ | $1.49 \times 10^{12}$ | $2.29 \times 10^{12}$ | $2.75 \times 10^{12}$ | 4.3 |
|  |  |  |  |  | 500 | $7.44 \times 10^{11}$ | $2.51 \times 10^{12}$ | $3.12 \times 10^{12}$ | $3.24 \times 10^{12}$ | 5.9 |
| CC-[1]$^+$ → CC-[2]$^{+,\ddagger}$ | 10.3 | −1.4 | 3.24 | 60 | 200 | $2.45 \times 10^{11}$ | $3.19 \times 10^{11}$ | $3.66 \times 10^{11}$ | $3.71 \times 10^{11}$ | 4.3 |
|  |  |  |  |  | 333 | $2.53 \times 10^{11}$ | $2.68 \times 10^{11}$ | $2.82 \times 10^{11}$ | $2.82 \times 10^{11}$ | 9.0 |
|  |  |  |  |  | 500 | $2.57 \times 10^{11}$ | $2.55 \times 10^{11}$ | $2.61 \times 10^{11}$ | $2.62 \times 10^{11}$ | 15.4 |
| CC-[2]$^{+,\ddagger}$ → CC-[3]$^+$ | 11.4 | −0.7 | 1.50 | 242 | 200 | $9.28 \times 10^{9}$ | $1.64 \times 10^{12}$ | $5.57 \times 10^{12}$ | — | 1.6 |
|  |  |  |  |  | 333 | $1.44 \times 10^{11}$ | $1.97 \times 10^{12}$ | $3.67 \times 10^{12}$ | $5.91 \times 10^{12}$ | 3.5 |
|  |  |  |  |  | 500 | $5.68 \times 10^{11}$ | $2.40 \times 10^{12}$ | $3.33 \times 10^{12}$ | $3.66 \times 10^{12}$ | 6.1 |
| CG-[1]$^0$ → CG-[2]$^{0,\ddagger}$ | 97.3 | 79.1 | 34.5 | 322 | 200 | $9.68 \times 10^{-15}$ | $1.42 \times 10^{-10}$ | $7.46 \times 10^{-10}$ | — | 86.0 |
|  |  |  |  |  | 333 | $1.47 \times 10^{-4}$ | $2.67 \times 10^{-2}$ | $6.77 \times 10^{-2}$ | $7.68 \times 10^{-2}$ | 92.0 |
|  |  |  |  |  | 500 | $1.81 \times 10^{1}$ | $3.85 \times 10^{2}$ | $6.47 \times 10^{2}$ | $8.66 \times 10^{2}$ | 99.9 |
| CG-[1]$^0$ → CG-[3]$^0$ | 191.2 | 176.9 | 76.6 | 18 | 200 | $3.41 \times 10^{-39}$ | $1.17 \times 10^{-35}$ | $1.18 \times 10^{-35}$ | $1.18 \times 10^{-35}$ | 182.1 |
|  |  |  |  |  | 333 | $3.26 \times 10^{-19}$ | $3.25 \times 10^{-17}$ | $3.27 \times 10^{-17}$ | $3.27 \times 10^{-17}$ | 187.2 |
|  |  |  |  |  | 500 | $3.18 \times 10^{-9}$ | $5.20 \times 10^{-8}$ | $5.21 \times 10^{-8}$ | $5.21 \times 10^{-8}$ | 194.3 |
| CC-[1]$^0$ → CC-[2]$^{0,\ddagger}$ | 88.7 | 64.8 | 29.0 | 274 | 200 | $3.93 \times 10^{-10}$ | $3.47 \times 10^{-5}$ | $1.42 \times 10^{-4}$ | — | 65.4 |
|  |  |  |  |  | 333 | $7.22 \times 10^{-1}$ | $2.97 \times 10^{2}$ | $6.27 \times 10^{2}$ | $1.45 \times 10^{3}$ | 66.2 |
|  |  |  |  |  | 500 | $3.10 \times 10^{4}$ | $9.82 \times 10^{5}$ | $1.47 \times 10^{6}$ | $1.71 \times 10^{6}$ | 67.3 |

(Continued.)

**Table 1.** (Continued.)

| channel | $\Delta E^{\ddagger}$ | $\Delta E^{\ddagger}_{ZPE}$ | $\Delta H^{\ddagger}$ | $T_c$ | $T$ | $k^{Class}_{f/r}$ | $k^{0\text{-vib}}_{f/r}$ | $k^{S\text{-Wig}}_{f/r}$ | $k^{F\text{-Wig}}_{f/r}$ | $\Delta G^{\ddagger/(Rel)}$ |
|---|---|---|---|---|---|---|---|---|---|---|
| CC-[1]$^0$ → CC-[3]$^0$ | 170.8 | 167.6 | 73.1 | 21 | 200 | $5.59 \times 10^{-33}$ | $1.75 \times 10^{-32}$ | $1.78 \times 10^{-32}$ | $1.78 \times 10^{-32}$ | 169.9 |
| | | | | | 333 | $3.71 \times 10^{-15}$ | $6.48 \times 10^{-15}$ | $6.52 \times 10^{-15}$ | $6.52 \times 10^{-15}$ | 172.4 |
| | | | | | 500 | $3.02 \times 10^{-6}$ | $4.08 \times 10^{-6}$ | $4.09 \times 10^{-6}$ | $4.09 \times 10^{-6}$ | 176.2 |
| CG-[1]$^{2+}$ → CG-[2]$^{2+,\ddagger}$ | 6.3 | −4.2 | −5.1 | 228 | 200 | $3.01 \times 10^{10}$ | $2.31 \times 10^{13}$ | $7.24 \times 10^{13}$ | — | (−2.9) |
| | | | | | 333 | $1.36 \times 10^{12}$ | $1.18 \times 10^{13}$ | $2.09 \times 10^{13}$ | $3.02 \times 10^{13}$ | (−1.5) |
| | | | | | 500 | $2.88 \times 10^{12}$ | $9.35 \times 10^{12}$ | $1.25 \times 10^{13}$ | $1.35 \times 10^{13}$ | 0.4 |
| CG-[2]$^{2+,\ddagger}$ ← CG-[3]$^{2+}$ | 19.0 | 6.7 | 5.5 | 228 | 200 | $6.27 \times 10^{7}$ | $1.61 \times 10^{10}$ | $5.03 \times 10^{10}$ | — | 9.2 |
| | | | | | 333 | $6.06 \times 10^{9}$ | $1.05 \times 10^{11}$ | $1.86 \times 10^{11}$ | $2.70 \times 10^{11}$ | 11.6 |
| | | | | | 500 | $5.96 \times 10^{10}$ | $2.94 \times 10^{11}$ | $3.95 \times 10^{11}$ | $4.25 \times 10^{11}$ | 14.8 |
| CC-[1]$^{2+}$ → CC-[2]$^{2+,\ddagger}$ | 64.7 | 39.2 | 19.2 | 240 | 200 | $9.09 \times 10^{-3}$ | $2.08 \times 10^{2}$ | $7.01 \times 10^{2}$ | — | 39.4 |
| | | | | | 333 | $5.28 \times 10^{4}$ | $7.08 \times 10^{6}$ | $1.32 \times 10^{7}$ | $2.09 \times 10^{7}$ | 38.2 |
| | | | | | 500 | $1.27 \times 10^{8}$ | $1.75 \times 10^{9}$ | $2.41 \times 10^{9}$ | $2.64 \times 10^{9}$ | 36.1 |
| CC-[2]$^{2+,\ddagger}$ ← CC-[3]$^{2+}$ | 67.7 | 44.2 | 20.8 | 238 | 200 | $1.03 \times 10^{-4}$ | $1.81 \times 10^{0}$ | $6.05 \times 10^{0}$ | — | 47.3 |
| | | | | | 333 | $1.22 \times 10^{3}$ | $1.62 \times 10^{5}$ | $2.99 \times 10^{5}$ | $4.67 \times 10^{5}$ | 48.7 |
| | | | | | 500 | $4.21 \times 10^{6}$ | $6.08 \times 10^{7}$ | $8.35 \times 10^{7}$ | $9.13 \times 10^{7}$ | 50.1 |

For $\varepsilon = 23$, a similar potential energy curve is shown in figure 2$b$, in which the same non-concerted, uphill bifunctional proton transfer with lower energy barriers and weaker H-bonds are observed; $\Delta E^{\ddagger}$, $\Delta E^{\text{H-bond/CP}}$ and $\Delta E^{\text{Solv}}$ for the transition structure **CC-[2]$^{0,\ddagger}$** are 63.3, $-232.1$ and $-140.8$ kJ mol$^{-1}$, respectively. The uphill bifunctional proton transfer for $\varepsilon = 1$ and 23 can be attributed to the strength of the ion-pair associations in the transition structures (**CG-[2]$^{0,\ddagger}$** and **CC-[2]$^{0,\ddagger}$**), which can be regarded as 'dipolar energy traps' [32]; because the ion-pair association in **CC-[2]$^{0,\ddagger}$** is strongly destabilized by the electric field of the local dielectric environment ($\varepsilon = 23$), $\Delta E^{\text{NSE}} = 324.6$ kJ mol$^{-1}$ (electronic supplementary material, table S2), the energy barrier is lower than that for $\varepsilon = 1$. Because the Grotthuss mechanism is characterized by the interconversion between precursor, transition state and proton transferred structures (e.g. the Eigen–Zundel–Eigen-like scenario) and bifunctional proton transfer in [PBI]$_2$ is governed by an uphill potential energy curve, the ion-pair transition structure of [PBI]$_2$ can be excluded from the Grotthuss mechanism for the PBI membrane.

For H$^+$[PBI]$_2$ with $\varepsilon = 1$, the potential energy curve in figure 3$a$ reveals that the interconversion between the precursor, transition state and proton transferred structures (**CG-[1]$^+$ $\rightleftharpoons$ CG-[2]$^{+,\ddagger}$ $\rightleftharpoons$ CG-[3]$^+$**) involves a low energy barrier; single-proton transfer in the forward direction is slightly more preferential (almost barrierless) than in the reverse direction, $\Delta E^{\ddagger} = 1.8$ and 12.8 kJ mol$^{-1}$, respectively. Due to the higher aromaticity in H$^+$[PBI]$_2$, $\Delta E^{\ddagger}$ is lower than that in H$^+$[Im]$_2$, $\Delta E^{\ddagger} = 9.5$ kJ mol$^{-1}$ [12]. $\Delta E^{\text{H-bond/CP}}$ for the transition structure **CG-[2]$^{+,\ddagger}$** is $-202.0$ kJ mol$^{-1}$, characterized by a shared proton in the H-bond (**1**) with $R_{\text{N–N}} = 2.60$, $\Delta d_{\text{DA}} = 0.14$ Å. The results are slightly different for $\varepsilon = 23$, in which the energy barriers for the interconversion (**CC-[1]$^+$ $\rightleftharpoons$ CC-[2]$^{+,\ddagger}$ $\rightleftharpoons$ CC-[3]$^+$**) are comparable in the forward and reverse directions, $\Delta E^{\ddagger} = 10.3$ and 11.4 kJ mol$^{-1}$, respectively. It should be noted that the destabilization effect of the protonated H-bond (**1**) in H$^+$[PBI]$_2$ is significantly less pronounced ($\Delta E^{\text{NSE}} = 39.0$ kJ mol$^{-1}$) than in [PBI]$_2$, resulting in only a slight change in the energy barrier.

For H$^{2+}$[PBI]$_2$, bifunctional proton transfer takes place preferentially and sequentially in $\varepsilon = 1$ (figure 4$a$). Interconversion (**CG-[1]$^{2+}$ $\rightleftharpoons$ CG-[2]$^{2+,\ddagger}$ $\rightleftharpoons$ CG-[3]$^{2+}$**) occurs on the low energy barrier path. The first transfer occurs in the H-bond (**1**) of the transition structure **CG-[2]$^{2+,\ddagger}$** (Zundel-like structure) with $\Delta E^{\ddagger} = 6.3$ kJ mol$^{-1}$ and $\Delta E^{\text{H-bond/CP}} = -160.0$ kJ mol$^{-1}$; similar shared proton structures were found in our previous studies as active intermediates in proton transfer in the O–H$^+$ $\cdots$ O H-bonds of H$^+$[H$_2$O]$_2$ and H$^+$[H$_3$PO$_4$]$_2$ for $\varepsilon = 1$ [21,33]. However, the second transfer involves an almost barrierless potential via a transition structure with $\Delta E^{\text{H-bond/CP}} \approx -269$ kJ mol$^{-1}$, leading to **CG-[3]$^{2+}$** with $\Delta E^{\text{H-bond/CP}} = -22.1$ kJ mol$^{-1}$. For $\varepsilon = 23$, the bifunctional proton transfer process is concerted with a higher energy barrier, $\Delta E^{\ddagger} = 64.7$ and 67.7 kJ mol$^{-1}$ in the forward and reverse directions, respectively. The increase in $\Delta E^{\ddagger}$ is in accordance with the H-bond interaction energy of **CC-[2]$^{2+,\ddagger}$**, $\Delta E^{\text{H-bond/CP}} = -207.2$ kJ mol$^{-1}$.

In conclusion, the role played by the local dielectric environment and the interplay among energies for the bifunctional proton transfer paths, the relative solvation ($\Delta E^{\text{Rel,Solv}}$), total ($\Delta E^{\text{Rel,Total}}$) and H-bond interaction ($\Delta E^{\text{Rel,H-bond/CP}}$) energies with respect to the precursors were computed and are included in figure 5. The trends for $\Delta E^{\text{Rel,Solv}}$ and $\Delta E^{\text{Rel,Total}}$ in figure 5$a$ suggest that for the protonated H-bond dimers (H$^{2+}$[PBI]$_2$ and H$^+$[PBI]$_2$), the increase in $\Delta E^{\ddagger}$ correlates with the increase in $\Delta E^{\text{Rel,Solv}}$ for the transition structure; the transition structure is destabilized compared with the precursors and the higher the number of protonated H-bonds, the higher the $\Delta E^{\ddagger}$ for $\varepsilon = 23$. By contrast, for [PBI]$_2$, the increase in $\Delta E^{\ddagger}$ is associated with the decrease in $\Delta E^{\text{Rel,Solv}}$; the positive and negative charges in the ion-pair transition structure are both increased (stabilized) compared with the precursor, and the lower $\Delta E^{\text{Rel,Solv}}$ is, the higher $\Delta E^{\ddagger}$ is for $\varepsilon = 23$. The latter prohibits the ion-pair transition structure from being involved in the Grotthuss mechanism.

The trends for $\Delta E^{\text{Rel,Solv}}$ and $\Delta E^{\text{Rel,H-bond/CP}}$ in figure 5$b$ yield the information in the same direction. For H$^{2+}$[PBI]$_2$ and H$^+$[PBI]$_2$, the increase in $\Delta E^{\text{Rel,Solv}}$ for the transition structure is correlated with the strength of the H-bonds; the stronger the H-bonds with respect to the precursor, the higher the $\Delta E^{\ddagger}$. This finding is in accordance with all of our previous studies, in which the rate-determining process for proton transfer is characterized by strong, protonated H-bonds with the oscillatory shuttling motion of the shared proton. For [PBI]$_2$ with $\varepsilon = 23$, the increase in the positive and negative charges along with the proton transfer path leads to strong electrostatic attraction in the ion-pair transition structure, which prohibits the Grotthuss mechanism; the Grotthuss mechanism takes place in dynamic H-bond networks, in which the formation and cleavage of covalent bonds are prerequisites. Based on the above discussion, one can anticipate that the rate-determining scenario for bifunctional proton transfer in the PBI membrane is most likely to be the concerted proton exchange in H$^{2+}$[PBI]$_2$, **CC-[1]$^{2+}$ $\rightleftharpoons$ CC-[2]$^{2+,\ddagger}$ $\rightleftharpoons$ CC-[3]$^{2+}$**. This piece of information was used as a guideline for calculation of the kinetics and proton conductivities.

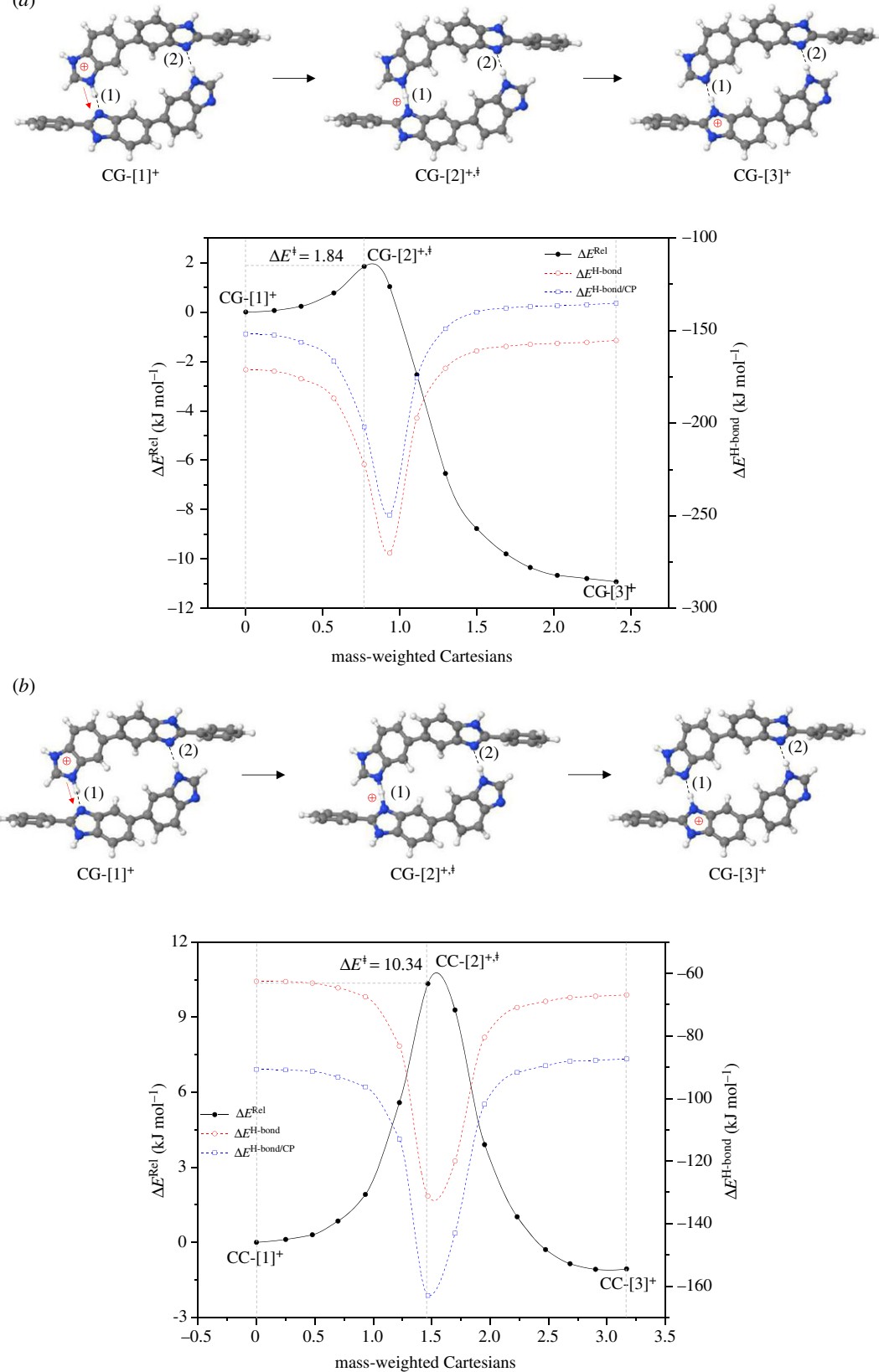

**Figure 3.** Reaction path and H-bond interaction energies for proton transfer in the single-protonated PBI dimer obtained based on the B3LYP/DZP and NEB methods. The three-character codes are explained in the text. (*a*)–(*b*) in the gas phase ($\varepsilon = 1$) and continuum dielectric environment ($\varepsilon = 23$), respectively; $\Delta E^{\text{Rel}}$ = relative total energy; $\Delta E^{\text{H-bond}}$ = H-bond interaction energy; $\Delta E^{\text{H-bond/CP}}$ = H-bond interaction energy obtained with the counterpoise correction of the BSSE; $\Delta E^{\ddagger}$ = energy barrier of proton transfer; x-axis = mass-weighted Cartesians.

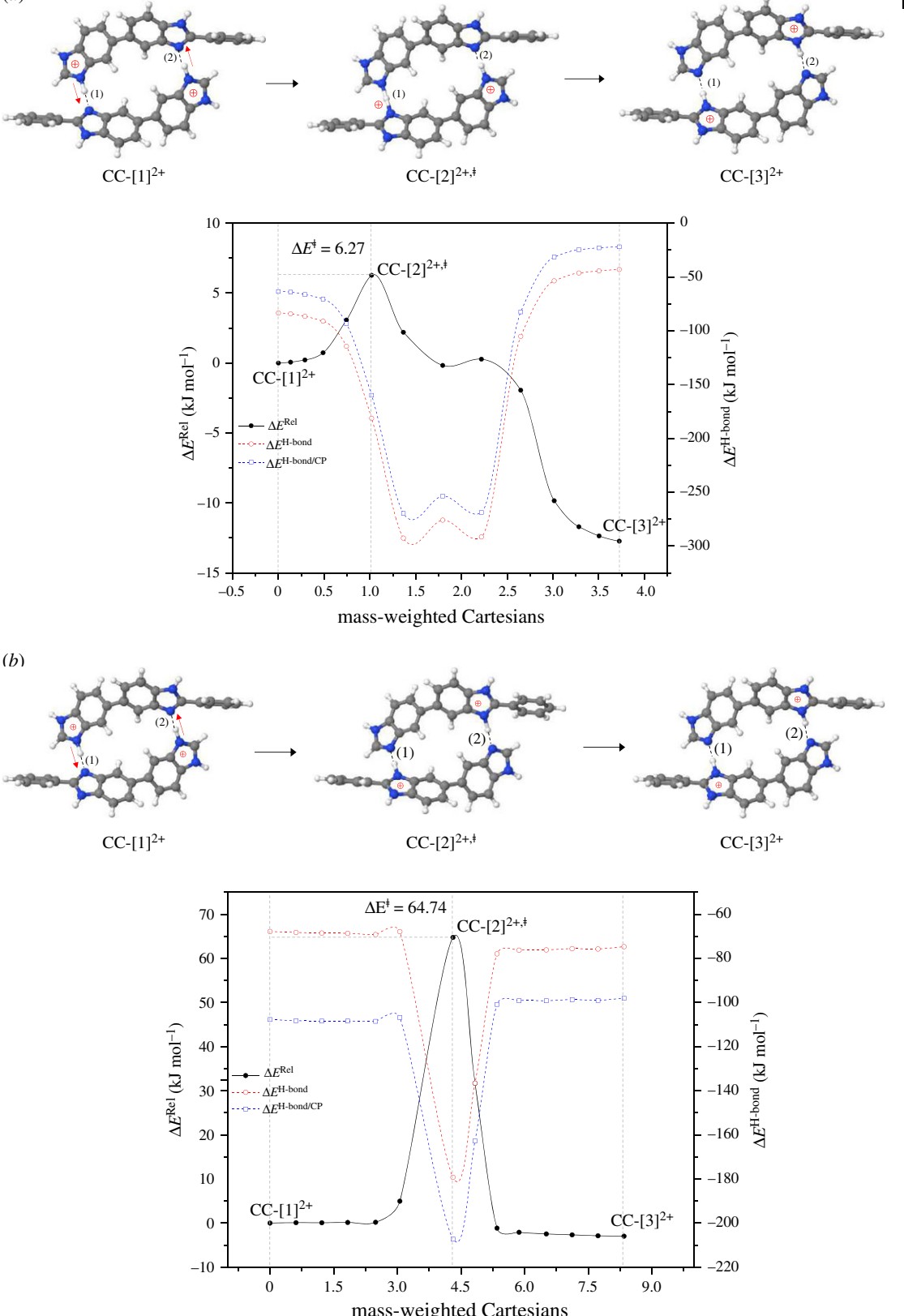

**Figure 4.** Reaction path and H-bond interaction energies for bifunctional proton transfer in the double-protonated PBI dimer obtained based on the B3LYP/DZP and NEB methods. The three-character codes are explained in the text. $(a)$–$(b)$ in the gas phase $(\varepsilon = 1)$ and continuum dielectric environment $(\varepsilon = 23)$, respectively; $\Delta E^{\text{Rel}}$ = relative energy; $\Delta E^{\text{H-bond}}$ = H-bond interaction energy; $\Delta E^{\text{H-bond/CP}}$ = H-bond interaction energy obtained with the counterpoise correction of the BSSE; $\Delta E^{\ddagger}$ = energy barrier of proton transfer; x-axis = mass-weighted Cartesians.

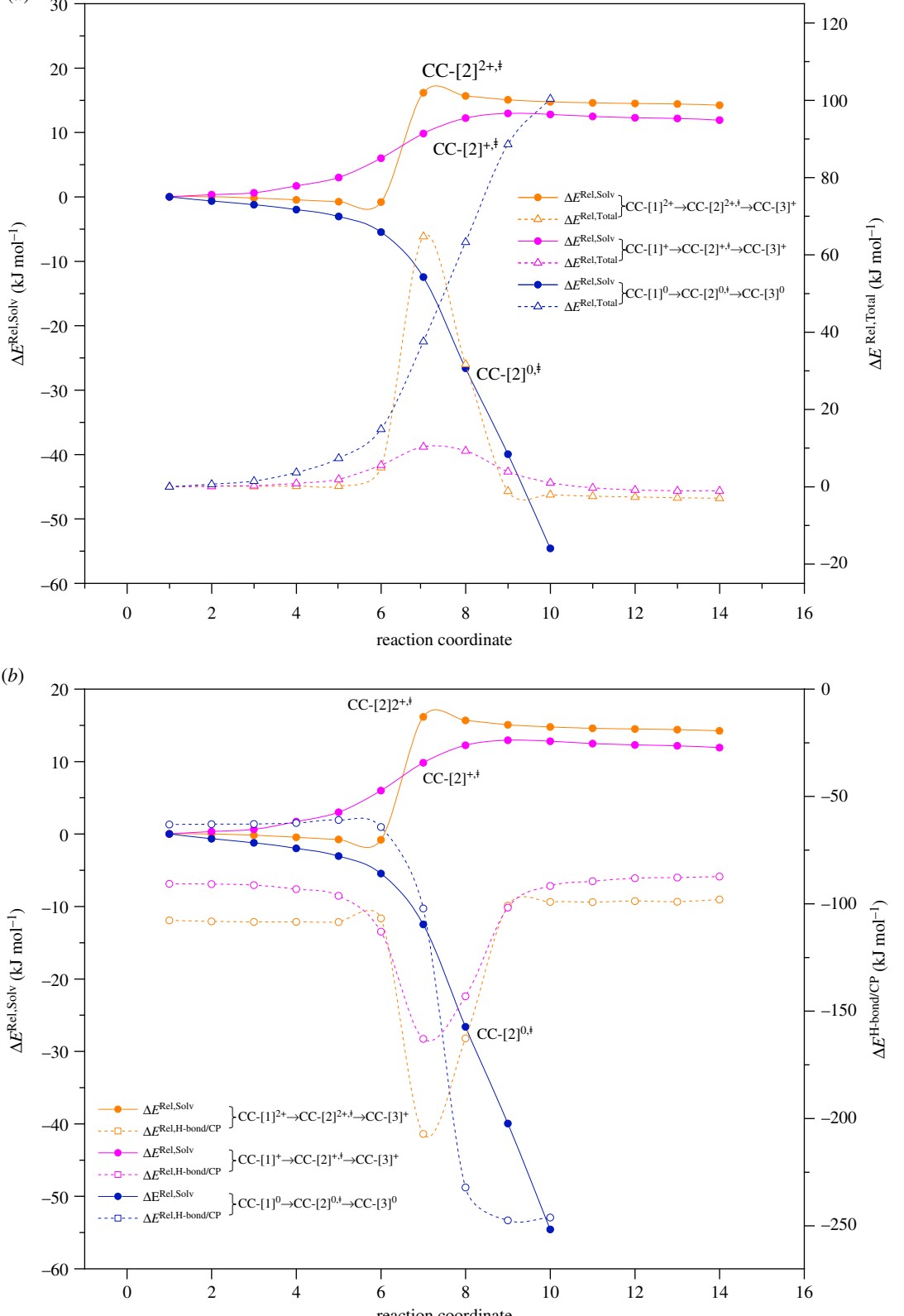

**Figure 5.** Plots of relative energies for the bifunctional proton transfer paths of $H^{2+}[PBI]_2$, $H^+[PBI]_2$, $[PBI]_2$ in high local dielectric environment obtained from the B3LYP/DZP method and counterpoise correction of BSSE. The relative energies are computed with respect to the corresponding precursors in figures 2–4 and bifunctional proton transfer paths were optimized using the NEB method. (*a*) Relative solvation energies ($\Delta E^{Rel,Solv}$) and relative total energies ($\Delta E^{Rel,Total}$). (*b*) Relative solvation energies ($\Delta E^{Rel,Solv}$) and relative H-bond interaction energies ($\Delta E^{Rel,H\text{-}bond/CP}$).

## 3.3. Kinetics of bifunctional proton transfer

The thermodynamic and kinetic results obtained based on TST are included in table 1. The temperature-dependent plots for the rate constants and quantum effect are included in the electronic supplementary material, figure S2 and shown as examples in figure 6. It appears in table 1 that the trends of the energy barriers without ($\Delta E^{\ddagger}$) and with the zero-point vibrational energy corrections ($\Delta E_{ZPE}^{\ddagger}$) are the same, for which are systematically lower. Analysis of the free energy barriers ($\Delta G^{\ddagger}$) in table 1 supports the discussion made based on $\Delta E^{\ddagger}$, for which the possibility for [PBI]$_2$ to be involved in the Grotthuss mechanism could be ruled out and formation of the transition structure (CC-[2]$^{2+,\ddagger}$) in the concerted bifunctional proton transfer in H$^{2+}$[PBI]$_2$ is thermodynamically less favourable in high local dielectric environment, e.g. at 333 K, $\Delta G^{\ddagger} = 38.2$ kJ mol$^{-1}$. Comparison of $\Delta G^{\ddagger}$ in $\varepsilon = 1$ and 23 also confirms the trends of the effect of local dielectric environment, for which the increase/decrease in $\Delta G^{\ddagger}$ in $\varepsilon = 23$ corresponds to the increase/decrease in the energy barriers ($\Delta E^{\ddagger}$ and $\Delta E_{ZPE}^{\ddagger}$) at the transition states. For example, for H$^{2+}$[PBI]$_2$ in $\varepsilon = 1$ and 23, $\Delta E^{\ddagger} = 6.3$ and 64.7 kJ mol$^{-1}$, and at 500 K, $\Delta G^{\ddagger} = 0.4$ and 36.1 kJ mol$^{-1}$, respectively.

The temperature dependence of $\Delta G^{\ddagger}$ appears to be the same for [PBI]$_2$ and H$^+$[PBI]$_2$, for which the increase in the temperature from 200 to 500 K leads to an increase in $\Delta G^{\ddagger}$. By contrast, for H$^{2+}$[PBI]$_2$ in $\varepsilon = 23$, the temperature increase leads to a decrease in $\Delta G^{\ddagger}$ from 39.4 to 36.1 kJ mol$^{-1}$. Analysis of the energetics in the electronic supplementary material, tables S1 and S2 suggests that these findings could be associated with the net stabilization/destabilization effect of local dielectric environment on the bifunctional H-bonds, especially for the transition structures; while $\Delta E^{NSE}$ for [PBI]$_2$ and H$^+$[PBI]$_2$ are all positive, the values for CC-[1]$^{2+}$ and CC-[2]$^{2+,\ddagger}$ are negatives, $\Delta E^{NSE} = -44.2$ and $-86.2$ kJ mol$^{-1}$, respectively. The latter reflects a strong stabilization effect of local dielectric environment on the double-protonated bifunctional H-bonds. The decrease in $\Delta G^{\ddagger}$ as the temperature increases could be attributed to the increase in the population of the shared proton N … H$^+$ … N H-bond; our BOMD simulations on the imidazole system (H$^+$[Im]$_n$, $n = 2$–4) [12] revealed that the intensity of the $^1$H NMR chemical shift associated with the shared proton (24 ppm) increases as the temperature increases from 298 to 500 K and the oscillatory shuttling motion of the shared proton possesses lower vibrational frequency than the asymmetric N-H$^+$ … N H-bond.

The values for the rate constants in table 1 indicate that the quantum effect is not negligible, except for [PBI]$_2$ with $\varepsilon = 1$ and 23 and H$^{2+}$[PBI]$_2$ with $\varepsilon = 23$, figure 6$a$ and $b$, respectively, for which all of the rate constants obey the Arrhenius equation. These bifunctional proton transfer scenarios will be discussed in comparison with available experimental data. In table 1, the classical rate constants ($k^{Class}$) are the lowest, as expected, whereas the rate constants with quantized vibrations ($k^{Q\text{-}vib}$) and with the second-order Wigner correction ($k^{S\text{-}Wig}$) are similar, especially at high temperatures. The temperature-dependent plots in figure 6 reveal that the rate constants with full Wigner correction ($k^{F\text{-}Wig}$) strongly deviate from linearity at low temperatures, indicating a strong quantum effect on bifunctional proton transfer, especially for H$^{2+}$[PBI]$_2$ in the forward direction for $\varepsilon = 1$ and for H$^+$[PBI]$_2$ for $\varepsilon = 1$ and 23, figure 6$b$ and $c$, respectively.

To complete the assessment of the quantum effect, the correlations of the rate constants obtained based on different methods are included in the electronic supplementary material, figure S3 and shown as examples in figure 7. The correlation plots suggest that for the rate-determining scenarios, the linear relationships between $k^{S\text{-}Wig}$ and $k^{Q\text{-}vib}$ exist over the studied temperature range with almost the same slopes and intercepts; for example, for H$^{2+}$[PBI]$_2$ in the forward direction with $\varepsilon = 23$ (CC-[1]$^{2+} \to$ CC-[2]$^{2+,\ddagger}$), $k_f^{S\text{-}Wig} = 0.95 \times k_f^{Q\text{-}vib} + 0.63$ (figure 7$a$) and for [PBI]$_2$ with $\varepsilon = 23$ (CC-[1]$^0 \to$ CC-[2]$^{0,\ddagger}$), $k_f^{S\text{-}Wig} = 0.96 \times k_f^{Q\text{-}vib} + 0.43$ (figure 7$b$). The correlation plots in figure 7 also show that for the rate-determining scenarios at temperatures above 350 K, linear relationships between $k^{F\text{-}Wig}$ and $k^{S\text{-}Wig}$ exist with approximately the same slopes and intercepts; for H$^{2+}$[PBI]$_2$ with $\varepsilon = 23$ (CC-[1]$^{2+} \to$ CC-[2]$^{2+,\ddagger}$), $k_f^{F\text{-}Wig} = 0.94 \times k_f^{S\text{-}Wig} + 0.61$ (figure 7$a$) and that for [PBI]$_2$ with $\varepsilon = 23$ (CC-[1]$^0 \to$ CC-[2]$^{0,\ddagger}$), $k_f^{F\text{-}Wig} = 0.93 \times k_f^{S\text{-}Wig} + 0.51$ (figure 7$b$). Because the quantum effect is not negligible and the trends for $k^{S\text{-}Wig}$ and $k^{F\text{-}Wig}$ are approximately the same above the operating temperatures of fuel cells (350 K or 1000/T = 2.86 K$^{-1}$), only $k^{S\text{-}Wig}$ will be used in further discussion.

For H$^{2+}$[PBI]$_2$ with $\varepsilon = 23$, the rate constants for bifunctional proton transfer at 333 K are $k_f^{S\text{-}Wig} = 1.32 \times 10^7$ and $k_r^{S\text{-}Wig} = 2.99 \times 10^5$ s$^{-1}$. These factors lead to equilibrium constants in the forward direction, $K_f^{S\text{-}Wig} = k_f^{S\text{-}Wig}/k_r^{S\text{-}Wig} = 4.40 \times 10^1$ (table 2), and in the reverse direction, $K_r^{S\text{-}Wig} = 1/K_f^{S\text{-}Wig} = 2.27 \times 10^{-2}$. It appears that at 300 K, the equilibrium constant for H$^{2+}$[PBI]$_2$ in the reverse direction is $K_r^{S\text{-Wig}} = 1.95 \times 10^{-2}$, which is similar to the experimental value of $3.6 \times 10^{-2}$ obtained based on the Scatchard method at 298 K [5]. Based on the following assumptions: (i) the acid dissociation constant is measured under acidic conditions; (ii) the steady state approximation occurs through the proton exchange reaction CC-[1]$^{2+} \rightleftharpoons$ CC-[2]$^{2+,\ddagger} \to$ CC-[3]$^{2+}$, in which the transition structure (CC-[2]$^{2+,\ddagger}$) is in

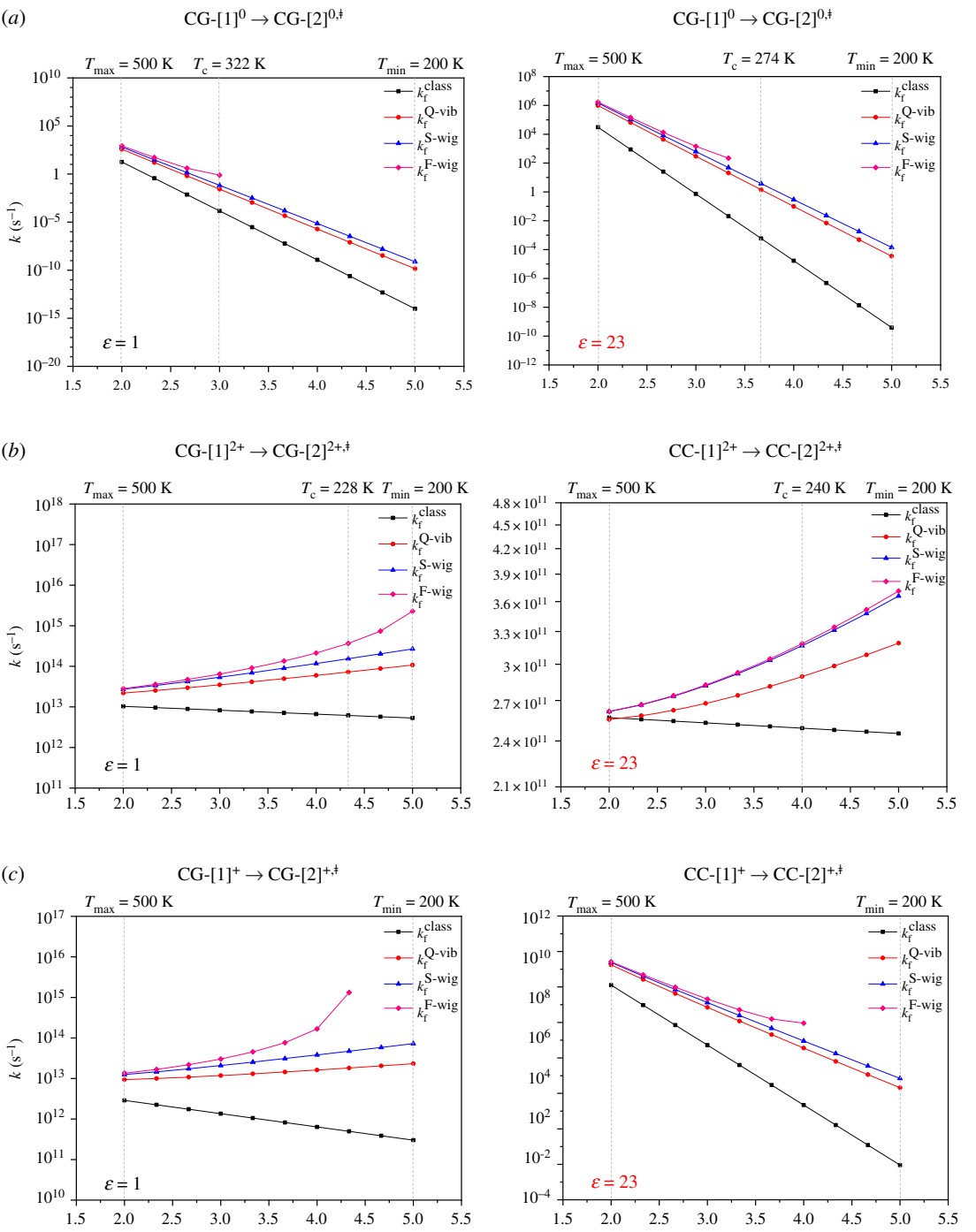

**Figure 6.** Plots of the rate constants for bifunctional proton transfer in the PBI dimers in the gas phase ($\varepsilon = 1$) and continuum dielectric environment ($\varepsilon = 23$) as a function of 1000/T obtained based on the TST. $k_f^{Class}$ = rate constant obtained from classical TST; $k_f^{Q\text{-}vib}$ = rate constant obtained with quantized vibrations including the zero-point vibrational energy; $k_f^{S\text{-}Wig}$ = rate constant obtained with quantized vibrations and tunnelling correction through the simplified Wigner correction to the second order; $k_f^{F\text{-}Wig}$ = full Wigner-corrected rate constant at T above $T_c$; f = forward direction. (a) Neutral dimer in the forward direction (**CG**-$[1]^0 \rightarrow$ **CG**-$[2]^{0,\ddagger}$ and **CC**-$[1]^0 \rightarrow$ **CC**-$[2]^{0,\ddagger}$). (b) Double-protonated dimer in the forward direction (**CG**-$[1]^{2+} \rightarrow$ **CG**-$[2]^{2+,\ddagger}$ and **CC**-$[1]^{2+} \rightarrow$ **CC**-$[2]^{2+,\ddagger}$). (c) Single-protonated dimer in the forward direction (**CG**-$[1]^+ \rightarrow$ **CG**-$[2]^{+,\ddagger}$ and **CC**-$[1]^+ \rightarrow$ **CC**-$[2]^{+,\ddagger}$).

equilibrium with the precursor (**CC**-$[1]^{2+}$); and (iii) the upper boundary for the rate constant for the barrierless potential can be roughly estimated to be $k = 10^{12}\,\text{s}^{-1}$ (the typical atomic vibrational period is 0.1–1.0 ps) [27], the acid dissociation constant ($K_a = k_f^{S\text{-}Wig}/k$) at 300 K is $2.46 \times 10^{-6}$ and $pK_a = 5.6$, which is in excellent agreement with the value obtained in the experiment ($pK_a \approx 5.5$) [1]. It should be noted that the experimental rate constants reported in the literature are based on different membrane

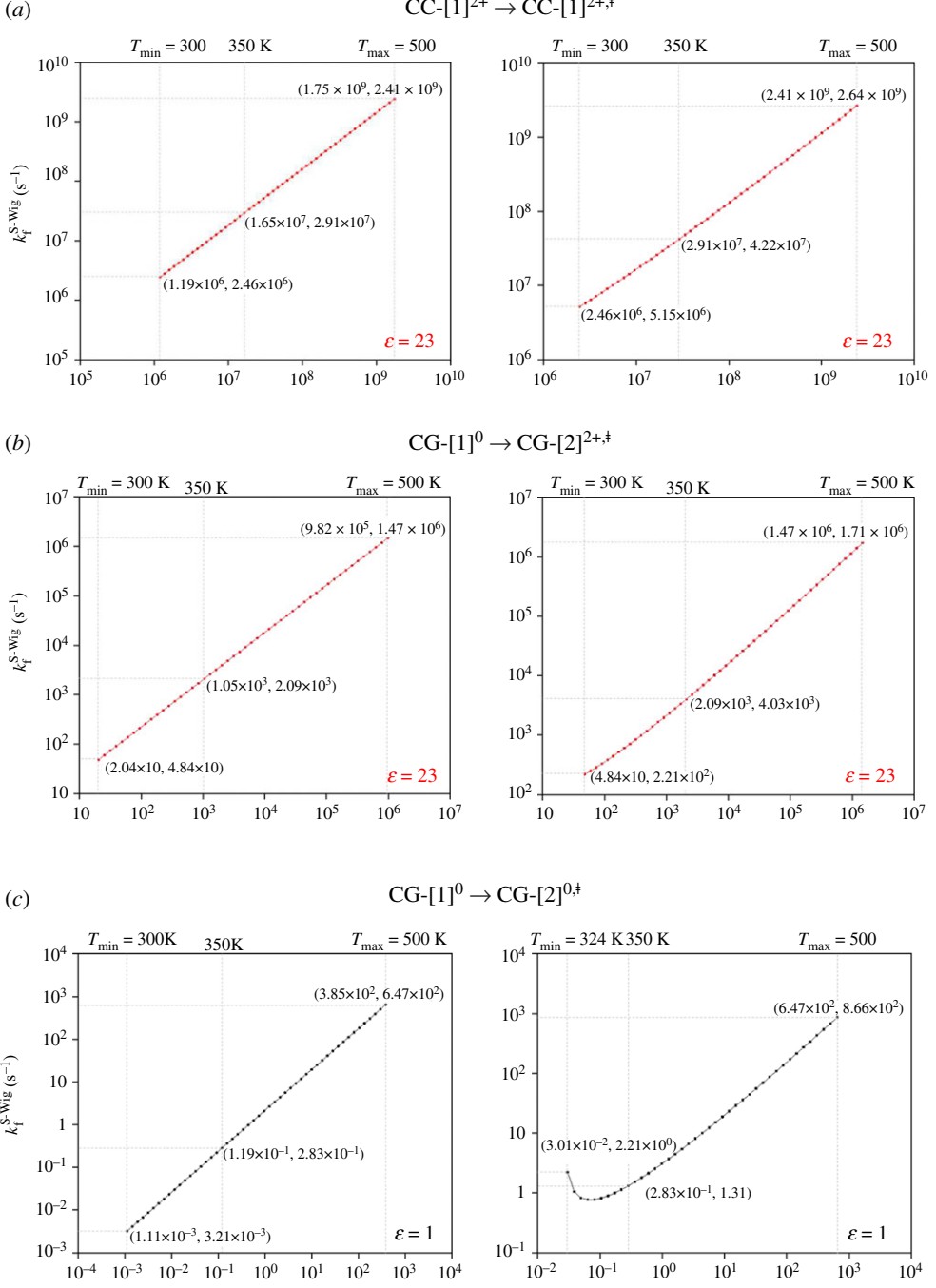

**Figure 7.** Correlations between the rate constants of the rate-determining processes obtained using different methods. $k_f^{Q\text{-vib}}$ = rate constant obtained with quantized vibrations including the zero-point vibrational energy; $k_f^{S\text{-Wig}}$ = rate constant obtained with quantized vibrations and tunnelling correction through the simplified Wigner correction to the second order; $k_f^{F\text{-Wig}}$ = full Wigner-corrected rate constant at $T$ above $T_c$; f = forward direction. (a) Double-protonated dimer in $\varepsilon = 23$ (**CC**-[1]$^{2+}$ → **CC**-[2]$^{2+,\ddagger}$). (b) Neutral dimer in $\varepsilon = 23$ (**CC**-[1]$^{0}$ → **CC**-[2]$^{0,\ddagger}$). (c) Neutral dimer in $\varepsilon = 1$ (**CG**-[1]$^{0}$ → **CG**-[2]$^{0,\ddagger}$).

preparation methods, measurement techniques and temperatures. Therefore, the agreement between the theoretical and experimental data is considered to be satisfactory.

## 3.4. Mobility of the phenyl groups

To prove the hypothesis of Fontanella *et al.* [10] that polymer segmental motion plays an important role in ion conductivity, potential energy curves for the torsional motion of the phenyl groups ($\omega_1$ and $\omega_2$ in figure 8a) were constructed for all of the dimers involved in the rate-determining processes.

**Table 2.** Equilibrium constants ($K$) obtained based on different rate constants ($k$) over the temperature range of 300–500 K. $K_f^{Class} = K$ obtained from $k_f^{Class}/k_r^{Class}$; $K_f^{Q\text{-vib}} = K$ obtained from $k_f^{Q\text{-vib}}/k_r^{Q\text{-vib}}$; $K_f^{S\text{-Wig}} = K$ obtained from $k_f^{S\text{-Wig}}/k_r^{S\text{-Wig}}$; $K_f^{F\text{-Wig}} = K$ obtained from $k_f^{F\text{-Wig}}/k_r^{F\text{-Wig}}$; f/r = forward or reverse direction.

| channel | $T$ | $K_f^{Class}$ | $K_f^{Q\text{-vib}}$ | $K_f^{S\text{-Wig}}$ | $K_f^{F\text{-Wig}}$ |
|---|---|---|---|---|---|
| CG-[1]$^+$ ⟵⟶ CG-[3]$^+$ | 200 | $7.16 \times 10^2$ | $1.69 \times 10^2$ | $1.69 \times 10^2$ | $1.69 \times 10^2$ |
| | 300 | $8.01 \times 10$ | $3.26 \times 10^1$ | $3.26 \times 10$ | $3.26 \times 10$ |
| | 333 | $5.16 \times 10$ | $2.34 \times 10$ | $2.34 \times 10$ | $2.34 \times 10$ |
| | 500 | $1.39 \times 10$ | $8.73 \times 10$ | $8.73 \times 10$ | $8.73 \times 10$ |
| CC-[1]$^+$ ⟵⟶ CC-[3]$^+$ | 200 | $2.64 \times 10$ | $1.95 \times 10^{-1}$ | $6.56 \times 10^{-2}$ | — |
| | 300 | $2.76 \times 10$ | $1.46 \times 10^{-1}$ | $7.51 \times 10^{-2}$ | $3.52 \times 10^{-2}$ |
| | 333 | $1.76 \times 10$ | $1.36 \times 10^{-1}$ | $7.68 \times 10^{-2}$ | $4.78 \times 10^{-2}$ |
| | 500 | $4.53 \times 10^{-1}$ | $1.06 \times 10^{-1}$ | $7.85 \times 10^{-2}$ | $7.15 \times 10^{-2}$ |
| CG-[1]$^{2+}$ ⟵⟶ CG-[3]$^{2+}$ | 200 | $4.79 \times 10^3$ | $1.44 \times 10^3$ | $1.44 \times 10^3$ | — |
| | 300 | $3.73 \times 10^2$ | $1.71 \times 10^2$ | $1.71 \times 10^2$ | $1.71 \times 10^2$ |
| | 333 | $2.24 \times 10^2$ | $1.12 \times 10^2$ | $1.12 \times 10^2$ | $1.12 \times 10^2$ |
| | 500 | $4.84 \times 10$ | $3.18 \times 10$ | $3.18 \times 10$ | $3.18 \times 10$ |
| CC-[1]$^{2+}$ ⟵⟶ CC-[3]$^{2+}$ | 200 | $8.83 \times 10$ | $1.14 \times 10^2$ | $1.16 \times 10^2$ | — |
| | 300 | $4.87 \times 10$ | $5.08 \times 10$ | $5.12 \times 10$ | $5.28 \times 10$ |
| | 333 | $4.32 \times 10$ | $4.36 \times 10$ | $4.40 \times 10$ | $4.48 \times 10$ |
| | 500 | $3.02 \times 10$ | $2.87 \times 10$ | $2.88 \times 10$ | $2.89 \times 10$ |
| CG-[1]$^+$ ⟵⟶ CG-[3]$^+$ | 200 | $7.16 \times 10^2$ | $1.69 \times 10^2$ | $1.69 \times 10^2$ | $1.69 \times 10^2$ |
| | 300 | $8.01 \times 10$ | $3.26 \times 10$ | $3.26 \times 10$ | $3.26 \times 10$ |
| | 333 | $5.16 \times 10$ | $2.34 \times 10$ | $2.34 \times 10$ | $2.34 \times 10$ |
| | 500 | $1.39 \times 10$ | $8.73$ | $8.73$ | $8.73$ |
| CC-[1]$^+$ ⟵⟶ CC-[3]$^+$ | 200 | $2.64 \times 10$ | $1.95 \times 10^{-1}$ | $6.56 \times 10^{-2}$ | — |
| | 300 | $2.76$ | $1.46 \times 10^{-1}$ | $7.51 \times 10^{-2}$ | $3.52 \times 10^{-2}$ |
| | 333 | $1.76$ | $1.36 \times 10^{-1}$ | $7.68 \times 10^{-2}$ | $4.78 \times 10^{-2}$ |
| | 500 | $4.53 \times 10^{-1}$ | $1.06 \times 10^{-1}$ | $7.85 \times 10^{-2}$ | $7.15 \times 10^{-2}$ |

The results show in general that within the thermal energy fluctuation at 333 K (RT = 2.77 kJ mol$^{-1}$, near the operating temperature of high-temperature fuel cells, approx. 350 K), $\omega_1$ and $\omega_2$ can be distorted by at most approximately ±14 degrees from the equilibrium structures (e.g. **CC-[1]$^{2+}$**) and at 500 K (RT = 4.16 kJ mol$^{-1}$), $\omega_1$ and $\omega_2$ can be distorted by at most approximately ±22 degrees. To study the effect of torsional motion, potential energy curves for bifunctional proton transfer in equilibrium structures with $\omega_2$ rotated by approximately 14 degrees were constructed; only $\omega_2$ was chosen because $\omega_1$ and $\omega_2$ in structure **CC-[3]$^{2+}$** are equivalent ($\omega_1$ and $\omega_2$ in figure 8b are 36.9 and 36.5 degrees, respectively).

The results show that all of the energy barriers are slightly lower, e.g. approximately 9–16 kJ mol$^{-1}$ lower for [PBI]$_2$, except for H$^{2+}$[PBI]$_2$ (the distorted **CC-[1]$^{2+}$** in figure 8b), for which $\Delta E^{\ddagger}$ reduces from 64.7 to 11.7 kJ mol$^{-1}$ in the forward direction and from 67.7 to 15.7 kJ mol$^{-1}$ in the reverse direction. This results in a considerable increase in the rate constants, for example, at 333 K, from $k_f^{S\text{-Wig}} = 1.32 \times 10^7$ s$^{-1}$ to $2.46 \times 10^{11}$ s$^{-1}$, and from $k_r^{S\text{-Wig}} = 2.99 \times 10^5$ s$^{-1}$ to $6.17 \times 10^9$ s$^{-1}$. Because the TST calculations suggest that the rate constant for torsional rotation in the distorted **CC-[1]$^{2+}$** is $k_f^{S\text{-Wig}} = 1.36 \times 10^{11}$ s$^{-1}$ at this temperature, polymer segmental motion could help promote the kinetics of ion transfer, as suggested by Fontanella *et al.* [10]. Although bifunctional proton transfer and torsional motions (polymer segmental motion) are coupled, bifunctional proton transfer in the distorted structure is faster than that in the equilibrium structure and does not to appear be the rate-determining process. This finding is in accordance with our previous study [12], in which reorientation of molecules in the H-bond network is not necessarily a key process.

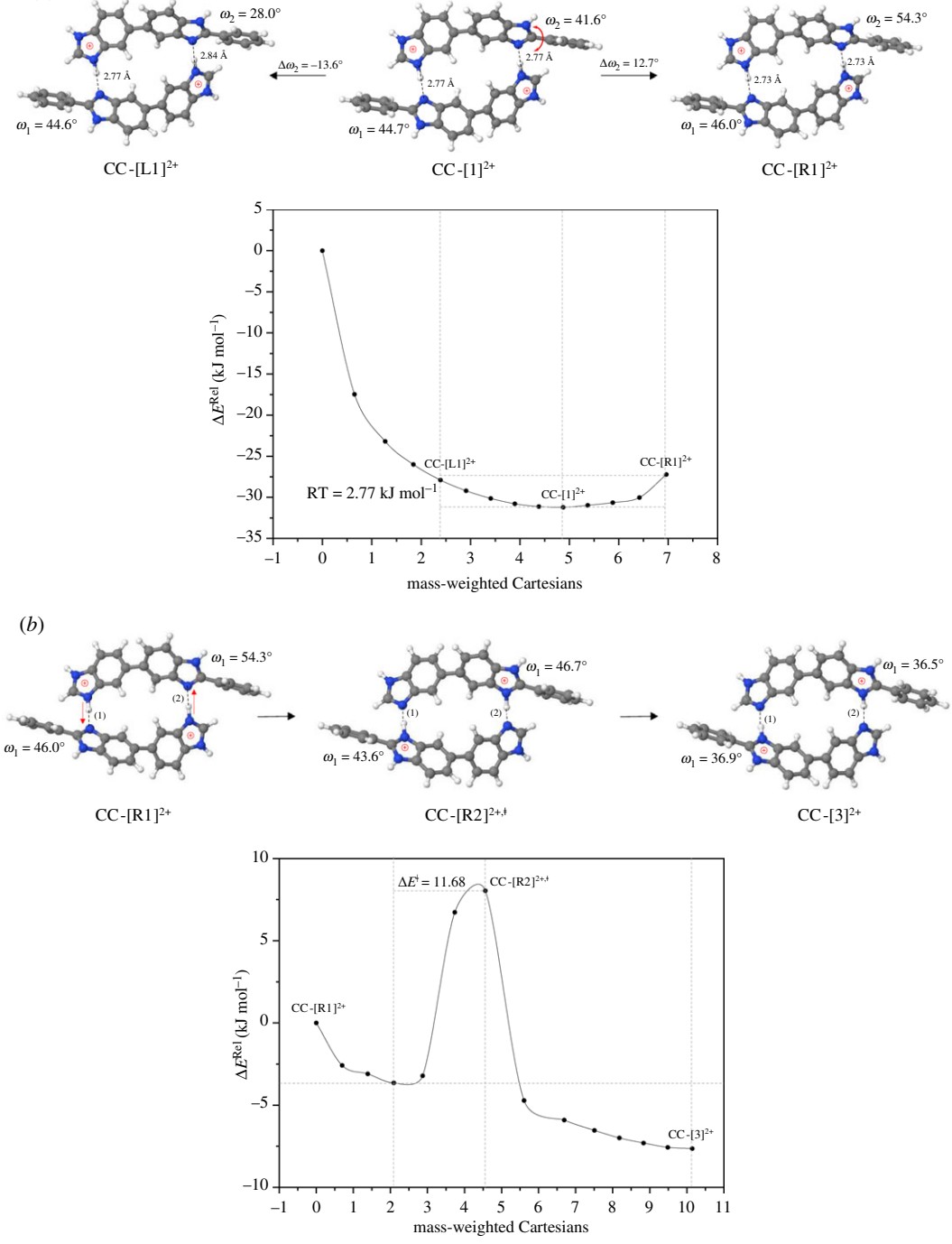

**Figure 8.** (*a*) potential energy curve for the torsional rotation ($\omega_2$) of the phenyl group in the double-protonated dimer in $\varepsilon = 23$, obtained from the B3LYP/DZP and NEB methods. Energies and angles are in kJ mol$^{-1}$ and in degree, respectively. $\Delta E^{Rel}$ = total energy with respect to structure **CC-**[1]$^{2+}$; **CC-[R1]**$^{2+}$ and **CC-[L1]**$^{2+}$ = rotated structures clockwise and counterclockwise, respectively. (*b*) Potential energy curve for bifunctional proton transfer in H-bonds of the distorted structure **CC-[R1]**$^{2+}$ obtained from the B3LYP/DZP and NEB methods. Energies and angles are in kJ mol$^{-1}$ and in degree, respectively. $\Delta E^{Rel}$ = total energy with respect to structure **CC-[R1]**$^{2+}$.

## 3.5. Proton conductivity

It should be noted that the reported experimental proton conductivities for pristine PBI are controversial. Proton conductivity values of $\sigma = 2$–$8 \times 10^{-4}$ S cm$^{-1}$ were at a relative humidity (RH) of 0–100% [19], whereas considerably lower values (approx. $10^{-12}$ S cm$^{-1}$) [2] were suggested to be more accurate, and a pure PBI membrane was concluded to be inapplicable as a solid electrolyte [16]. In this work, based on the pre-exponential factor determined using conductivity measurements in [9] ($A = 1.0 \times$

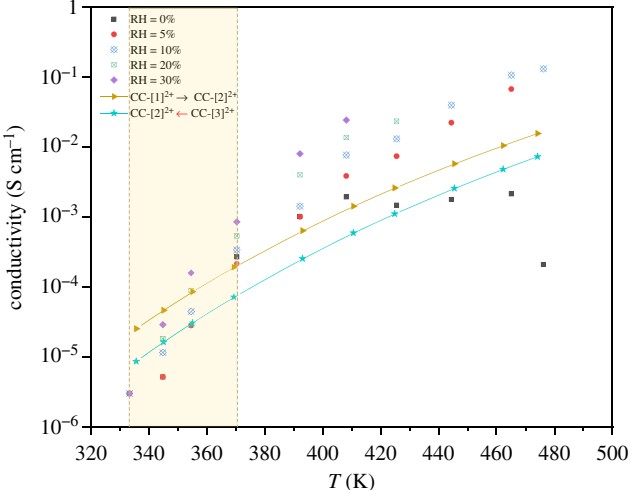

**Figure 9.** Plots of the conductivities ($\sigma$) of acid-doped PBI membranes as a function of temperature obtained from experiments (300% doping level and RH from 0–30%) [6], compared with those of the double-protonated dimer (**CC**-[1]$^{2+}$ → **CC**-[2]$^{2+,\ddagger}$ and **CC**-[2]$^{2+,\ddagger}$ ← **CC**-[3]$^{2+}$). Conductivities and temperatures are in S cm$^{-1}$ and K, respectively. RH = relative humidity.

$10^8$ S K cm$^{-1}$), the proton conductivities in [PBI]$_2$ for $\varepsilon = 1$ are $4$–$9 \times 10^{-12}$ S cm$^{-1}$ over the temperature range of 300–308 K, which are comparable with the results in [2]. Whereas the values for $\varepsilon = 23$, $\sigma = 1$–$7 \times 10^{-4}$ S cm$^{-1}$ over the temperature range of 500–550 K are compatible with those reported in [19]; our conductivity at 445 K ($\sigma = 10^{-5}$ S cm$^{-1}$) is in excellent agreement with the value reported for the H$_3$PO$_4$-doped PBI membrane at 433 K in [7], $\sigma \approx 10^{-5}$ S cm$^{-1}$. Based on the above discussion, one can anticipate that the discrepancies between the values in [2], [7] and [19] result from the difference in the membrane preparation and conductivity measurement techniques used; e.g. the higher values in [19] were obtained using so-called 'protodes', for which the conductivities are measured through the reaction between H$^+$ and e$^-$, Pt | H$_2 \rightleftharpoons$ 2H$^+$ (in PBI membrane) + 2e$^-$ (in wire), whereas the values reported in [2] were measured in an acid solution.

Because bifunctional proton transfer in H$^{2+}$[PBI]$_2$ can represent the rate-determining scenario in acid-doped PBI membranes, the temperature dependence of the proton conductivity through the elementary processes **CC**-[1]$^{2+}$ → **CC**-[2]$^{2+,\ddagger}$ and **CC**-[2]$^{2+,\ddagger}$ ← **CC**-[3]$^{2+}$ was studied over the temperature range of 200–500 K. Based on the energy barriers in table 1 and the pre-exponential term [9], $A = 1.0 \times 10^8$ S K cm$^{-1}$, proton conductivities were computed and compared with the experimental values obtained for the acid-doped PBI membranes (doping level of 300% and RH ranging from 0 to 30%) over the temperature range of 330–470 K [6] in figure 9. It appears that our theoretical results, which are based on the Grotthuss mechanism, agree reasonably well over the temperature range of 330–370 K ($10^{-4}$–$10^{-5}$ S cm$^{-1}$), above which the experimental proton conductivities are higher, especially for RH > 0%; proton conduction in acid-doped PBI membranes was suggested to result from the Grotthuss mechanism only at temperatures below the boiling point of water (373 K) [34]. It should be noted that because the conductivities measured in experiments depend upon experimental conditions (pH, RH, doping level, etc.), the agreement between the theoretical and experimental values is considered again to be satisfactory.

## 4. Conclusion

In this work, the Grotthuss mechanism for bifunctional proton transfer in PBI membranes was studied using the B3LYP/DZP and B3LYP/TZP methods and transition state theory (TST). This study used [PBI]$_2$, H$^+$[PBI]$_2$ and H$^{2+}$[PBI]$_2$ as model systems, in which the bifunctional proton transfer paths in low ($\varepsilon = 1$) and high ($\varepsilon = 23$) local dielectric environments were optimized using the NEB method. From analysis of the equilibrium structures, relative H-bond interaction energies and energy barriers showed that the B3LYP/DZP and B3LYP/TZP methods yield approximately the same results.

It appeared that the potential energy curves and strength of the H-bonds are sensitive to the local dielectric environment, and the uphill bifunctional proton transfer in [PBI]$_2$ for $\varepsilon = 1$ can be attributed to extraordinarily strong ion-pair H-bonds in the transition structure, regarded as a 'dipolar energy trap'. A high local dielectric environment ($\varepsilon = 23$) stabilizes [PBI]$_2$ but destabilizes the ion-pair in the transition structure, leading to a decrease in the energy barrier. Because the bifunctional proton

transfer paths for [PBI]$_2$ involve uphill potential energy curves and the Grotthuss mechanism generally takes place in dynamic H-bond networks, in which the formation and cleavage of covalent bonds are the fundamental steps, a dipolar energy trap can prohibit interconversion between the precursor and proton transferred structures, which rules out the possibility for [PBI]$_2$ to be involved in the Grotthuss mechanism in the PBI membrane.

Different results were observed for H$^+$[PBI]$_2$ and H$^{2+}$[PBI]$_2$, in which the increase in the energy barrier for $\varepsilon = 23$ is due to the stabilization of the protonated H-bonds in the transition structure compared with the precursor; while the protonated H-bonds in the transition structure are stronger than those in the precursor, the solute-local dielectric environment interaction is weaker, especially for H$^{2+}$[PBI]$_2$. Because the effect is considerably stronger than that of H$^+$[PBI]$_2$, one can anticipate that the rate-determining scenario for bifunctional proton transfer in the PBI membrane is most likely characterized by concerted proton exchanges in H$^{2+}$[PBI]$_2$, **CC**-[1]$^{2+}$ ⇌ **CC**-[2]$^{2+,\ddagger}$ ⇌ **CC**-[3]$^{2+}$; although the segmental (torsional) motion of the PBI chain can promote bifunctional proton transfer by lowering the energy barrier, the coupled motion of proton transfer and torsional rotation has a relatively high rate constant and does not appear to be the rate-determining process. Analysis of the rate constants obtained based on different approximations over the temperature range of 200–500 K confirmed that the quantum effect is not negligible for N–H$^+$…N H-bonds, especially at low temperatures, and the rates for concerted bifunctional proton transfer in H$^{2+}$[PBI]$_2$ in a high local dielectric environment obey the Arrhenius equation.

An attempt was made to correlate the present results with experimental data, especially for bifunctional proton transfer pathways that obey the Arrhenius equation. Based on the rate constants obtained with the second-order Wigner correction, the equilibrium constant for bifunctional proton transfer in H$^{2+}$[PBI]$_2$ in a high local dielectric environment is similar to the experimental acid dissociation constant obtained using the Scatchard method, and the $pK_a$ obtained based on the steady state approximation is in excellent agreement with the reported experimental value. Based on the energy barriers and experimental pre-exponential term, the temperature dependence of the proton conductivity was studied and compared with the experimental data for the acid-doped PBI membranes. It appears that the computed conductivities agree well with the experimental values over the temperature range of 330–370 K; the temperature range over which the proton conductivity obeys the Arrhenius equation. Finally, because the rate-determining scenario for the Grotthuss mechanism involves proton exchange in a high local dielectric environment, a low local dielectric environment can be one of the necessary conditions for effective bifunctional proton transfer in acid-doped PBI membranes. These theoretical results provide insights into the Grotthuss mechanism, which can be used as guidelines for understanding the fundamentals of proton transfers in other bifunctional H-bond systems.

Data accessibility. The data are provided in the electronic supplementary material [35].

Authors' contributions. All authors gave final approval for publication and agreed to be held accountable for the work performed therein.

Competing interests. We declare we have no competing interests.

Funding. This work was supported by Suranaree University of Technology (SUT) and by Office of the Higher Education Commission under NRU Project of Thailand. The financial support provided by the Thailand Research Fund (TRF) (grant no. MRG6180120) to J.T. is gratefully acknowledged.

Acknowledgements. P.P. would like to express sincere thanks to the Kittibandit Scholarship of SUT. The high-performance computer facilities provided by the following organizations are gratefully acknowledged: Institute of Science, SUT, National e-Science project of the National Electronics and Computer Technology Centre (NECTEC) and the National Science and Technology Development Agency (NSTDA).

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
