## [Peer Review File · Royal Society Open Science]

Review History

RSOS-211168.R0 (Original submission)

Review form: Reviewer 1

Is the manuscript scientifically sound in its present form?

Yes

Are the interpretations and conclusions justified by the results?

Yes

Is the language acceptable?

Yes

Do you have any ethical concerns with this paper?

No

Have you any concerns about statistical analyses in this paper?

No

Recommendation?

Accept with minor revision (please list in comments)

Comments to the Author(s)

See attached file (Appendix A).

Review form: Reviewer 2**Is the manuscript scientifically sound in its present form?**

Yes

Are the interpretations and conclusions justified by the results?

Yes

Is the language acceptable?

Yes

Do you have any ethical concerns with this paper?

No

Have you any concerns about statistical analyses in this paper?

No

Recommendation?

Major revision is needed (please make suggestions in comments)

Comments to the Author(s)

This is nice study to study the mechanism of proton transfer in high-temperature polymer electrolyte membrane for fuel cells. They could determine the different pathways, stability of transition states and kinetics of reactions of bifunctional proton transfer in poly(benzimidazole) under their neutral and cationic forms.

What are the real-life conditions where the values of the dielectric media can be 1 and 23? How is the conversion of the lowest and highest effects of the electric fields translated to dielectric constants of 1 and 23?

In the methodology: "for which the effect of the size was systematically investigated using the B3LYP/TZP method" it is not clear what is meant by the size. Also, what is the limit of the size where the double zeta vs the triple zeta basis sets are used?

Which double zeta and which triple zeta basis sets are used? Are they Pople, Dunning, etc.? Do they have diffuse functions, etc.?

Why is B3LYP used as method? It does not account for dispersion which is important in the studied systems.

How can the results of this study be used practically? What is suggested to use as material for fuel cells? And are the suggested systems feasible to be implemented in existing fuel cells?

Legends in Figure 6 are too tiny to be seen, they need to be adjusted.

"which can be used as a guideline for future experiments and the development of the same and similar systems" in the conclusion: very vague, needs more specifics.

The conclusion is very long, it needs to be more concise.

Decision letter (RSOS-211168.R0)

Dear Professor Sagarik:

Title: The Grotthuss mechanism for bifunctional proton transfer in poly(benzimidazole)
Manuscript ID: RSOS-211168

Thank you for submitting the above manuscript to Royal Society Open Science. On behalf of the Editors and the Royal Society of Chemistry, I am pleased to inform you that your manuscript will be accepted for publication in Royal Society Open Science subject to minor revision in accordance with the referee suggestions. Please find the reviewers' comments at the end of this email.

The reviewers and handling editors have recommended publication, but also suggest some minor revisions to your manuscript. Therefore, I invite you to respond to the comments and revise your manuscript.

Because the schedule for publication is very tight, it is a condition of publication that you submit the revised version of your manuscript before 27-Oct-2021. Please note that the revision deadline will expire at 00.00am on this date. If you do not think you will be able to meet this date please let me know immediately.

Kind regards,
Dr Ellis Wilde
Publishing Editor, Journals

On behalf of the Subject Editor Professor Anthony Stace and the Associate Editor Professor Kim Jelfs.

RSC Associate Editor
Comments to the Author:
(There are no comments.)

RSC Subject Editor
Comments to the Author:
(There are no comments.)

Reviewer comments to Author:
Reviewer: 1
Comments to the Author(s)
See attached file

Reviewer: 2
Comments to the Author(s)
This is nice study to study the mechanism of proton transfer in high-temperature polymer electrolyte membrane for fuel cells. They could determine the different pathways, stability of transition states and kinetics of reactions of bifunctional proton transfer in poly(benzimidazole) under their neutral and cationic forms.
What are the real-life conditions where the values of the dielectric media can be 1 and 23? How is the conversion of the lowest and highest effects of the electric fields translated to dielectric constants of 1 and 23?

In the methodology: “for which the effect of the size was systematically investigated using the B3LYP/TZP method” it is not clear what is meant by the size. Also, what is the limit of the size where the double zeta vs the triple zeta basis sets are used?

Which double zeta and which triple zeta basis sets are used? Are they Pople, Dunning, etc.? Do they have diffuse functions, etc.?

Why is B3LYP used as method? It does not account for dispersion which is important in the studied systems.

How can the results of this study be used practically? What is suggested to use as material for fuel cells? And are the suggested systems feasible to be implemented in existing fuel cells?

Legends in Figure 6 are too tiny to be seen, they need to be adjusted.

“which can be used as a guideline for future experiments and the development of the same and similar systems” in the conclusion: very vague, needs more specifics.

The conclusion is very long, it needs to be more concise.

Author's Response to Decision Letter for (RSOS-211168.R0)

See Appendix B.

RSOS-211168.R1 (Revision)

Review form: Reviewer 1

Is the manuscript scientifically sound in its present form?

Yes

Are the interpretations and conclusions justified by the results?

Yes

Is the language acceptable?

Yes

Do you have any ethical concerns with this paper?

Yes

Have you any concerns about statistical analyses in this paper?

No

Recommendation?

Accept as is

Comments to the Author(s)

The authors have addressed all the issues. In my opinion the manuscript can be published in the present form.

Decision letter (RSOS-211168.R1)

Dear Professor Sagarik:

Title: The Grotthuss mechanism for bifunctional proton transfer in poly(benzimidazole)
Manuscript ID: RSOS-211168.R1

It is a pleasure to accept your manuscript in its current form for publication in Royal Society Open Science. The chemistry content of Royal Society Open Science is published in collaboration with the Royal Society of Chemistry.

Yours sincerely,
Dr Ellis Wilde
Publishing Editor, Journals

On behalf of the Subject Editor Professor Anthony Stace and the Associate Editor Professor Kim Jelfs.

RSC Associate Editor
Comments to the Author:
(There are no comments.)

RSC Subject Editor
Comments to the Author:
(There are no comments.)

Reviewer(s)' Comments to Author:
Reviewer: 1

Comments to the Author(s)

The authors have addressed all the issues. In my opinion the manuscript can be published in the present form.

Appendix A

The manuscript “The Grotthuss mechanism for bifunctional proton transfer in poly(benzimidazole)” by Sagarik and coworkers reports a computational study, based on quantum-chemical calculations followed by kinetic simulations in the framework of Transition State Theory, aimed at elucidating the energetic and kinetic features of the bifunctional proton transfer in Poly(benzimidazole) (PBI) dimers utilized as models for mimicking the possible events underlying the high-temperature polymer electrolyte membranes for fuel cells. In particular the authors make use of three different models, i.e. $[PBI]_2$, $H+[PBI]_2$ and $H_2+[PBI]_2$. According to their results, the authors suggest that that: $[PBI]_2$ can be excluded by the Grotthuss mechanism, whereas for $H+[PBI]_2$ and $H_2+[PBI]_2$ an environmental effect is supposed to be relevant for the height of the related kinetic barrier; moreover a not negligible quantum effect is observed for the proton-transfer kinetics at lower temperatures. The presented work is scientifically sound, interesting and relatively well written. I recommend its publication even though, in my opinion, some minor points probably need to be addressed.

1) p. 4 “with the Lee–Yang–Parr (B3LYP) hybrid functional” should be integrated also by the meaning of “B”, i.e. the Becke’s term in the exchange-correlation functional linearly-combined equation.

2) p. 7. “The strength of the PBI-local dielectric environment interaction was approximated using the solvation energy (ΔE_{Solv}), computed from the difference between the total energies (E_{Total}) obtained with and without COSMO”;

in this respect I would like to remark that this is not fully coherent by a physical point of view. I am well aware of the fact that essentially this is what is carried out by many of the investigators dealing with condensed-phase calculations and, hence, I do not consider such a point as a flaw of this study. At the same time it is my duty to point out that dielectric constant is a temperature-dependent quantity and, hence, it should be always compared to free-energy differences and not to electronic-energies (i.e. without any thermal contribution) differences. Much more correct – but also much more complicated – would be the use of the electric field produced by the environment and the related fluctuations.

3) Probably because of my ignorance, it is rather obscure to me why the authors calculate the rate coefficients using the semiclassical limit (eq. 1) whose physical consistency is rather low (and I’m optimistic!!). In other words, what should be done in principle, it is to divide the internal degrees of freedom (dof) of the reactant and transition structure between semi-classical dof, e.g. internal rotations (see also the section concerning the phenyl ring rotations), and quantum dof.

Subsequently, on this basis, one should calculate the partition function of the classical dof with a phase integral and using the standard formulations used by the authors in eq. 2 for the latter one. This is very complicated and eq. 2 is certainly adequate for the purposes of the authors. But what about eq. 1? Since some of the readers, like me, might have this doubt it would be good to insert a small clarifying sentence or a supporting note.

4) I hardly understand the meaning of the physical – not numerical!! – effect of the rate coefficients (a non-equilibrium property strongly related to relaxation phenomena) on the equilibrium constants (by definition independent from any relaxation or time dependent phenomena). I would expect, as already remarked in the previous point, that the equilibrium constant can be affected by the way in which the partition function (i.e. the free energies) are evaluated but, frankly, I need a further explanation for understanding what tunneling has to do with free energy. Rather, I would say that the good agreement observed by the authors in this respect is due to the good level of their calculations.

5) p. 14 “The results show in general that within the thermal energy fluctuation at room temperature ($RT = 2.77$ kJ/mol).....”. I agree with such a statement...but the authors are spanning temperatures rather different from the room temperature. Therefore this section is rather fuzzy and incoherent or, at least, incomplete.

Appendix B

Comment/Response Summary

ID: RSOS-211168

Title: The Grotthuss mechanism for bifunctional proton transfer in poly(benzimidazole)

The authors would like to express sincere thanks to the editor and reviewers of “Royal Society Open Science” for valuable suggestions and comments on the research manuscript entitled “The Grotthuss mechanism for bifunctional proton transfer in poly(benzimidazole)” (RSOS-211168).

We seriously considered the comments and suggestions and revised the manuscript accordingly. In the revised manuscript and this comment/response summary, “blue fonts” highlight the texts that are modified/included. In this comment/response summary, the page and line numbers in the revised manuscript are cited to locate the places where the texts are modified or included.

Reviewer: 1

The manuscript “The Grotthuss mechanism for bifunctional proton transfer in poly(benzimidazole)” by Sagarik and coworkers reports a computational study, based on quantum chemical calculations followed by kinetic simulations in the framework of Transition State Theory, aimed at elucidating the energetic and kinetic features of the bifunctional proton transfer in Poly(benzimidazole) (PBI) dimers utilized as models for mimicking the possible events underlying the high-temperature polymer electrolyte membranes for fuel cells.

In particular, the authors make use of three different models, i.e. [PBI]₂, H+[PBI]₂ and H₂+[PBI]₂. According to their results, the authors suggest that that: [PBI]₂ can be excluded by the Grotthuss mechanism, whereas for H+[PBI]₂ and H₂+[PBI]₂ an environmental effect is supposed to be relevant for the height of the related kinetic barrier; moreover, a not negligible quantum effect is observed for the proton-transfer kinetics at lower temperatures. The presented work is scientifically sound, interesting and relatively well written.

I recommend its publication even though, in my opinion, some minor points probably need to be addressed.

Comment 1:

1) p. 4 “with the Lee–Yang–Parr (B3LYP) hybrid functional” should be integrated also by the meaning of “B”, i.e. the Becke’s term in the exchange-correlation functional linearly-combined equation.

Response 1:

Page 4, Line 22:

“the Lee–Yang–Parr (B3LYP)” is replaced by “the Becke,-3 Parameter, Lee-Yang-Parr (B3LYP)”.

Comment 2:

2) p. 7. “The strength of the PBI-local dielectric environment interaction was approximated using the solvation energy (ΔE_{Solv}), computed from the difference between the total energies (E_{Total}) obtained with and without COSMO”; in this respect I would like to remark that this is not fully coherent by a physical point of view. I am well aware of the fact that essentially this is what is carried out by many of the investigators dealing with condensed-phase calculations and, hence, I do not consider such a point as a flaw of this study.

At the same time, it is my duty to point out that dielectric constant is a temperature-dependent quantity and, hence, it should be always compared to free-energy differences and not to electronic energies (i.e. without any thermal contribution) differences. Much more correct – but also much more complicated – would be the use of the electric field produced by the environment and the related fluctuations.

Response 2:

In this study, we would like to investigate the role played by local dielectric environment surrounding the bifunctional H-bonds, focusing on how the energy barriers (potential energy curves), H-bond interaction energies and rate constants change without and with local dielectric environment. We are aware of the fact that the dielectric constant is temperature dependent; an experiment suggested that for cyclohexanone, ϵ varies from 15 to 12 over the temperature range of 293-343 K (<https://doi.org/10.1016/j.spjpm.2015.03.008>). However, because we do not know the exact value of the dielectric constant at/in the vicinity of the bifunctional H-bonds (which could be different from the bulk) and range over which it fluctuates at a particular temperature, we decided to study the effect using the lowest possible ($\epsilon = 1$ in the gas phase) and average ($\epsilon = 23$ in the bulk) values; the thermal energy agitations lead to fluctuations of the dipole orientations and dielectric environment surrounding molecules and it is difficult to approximate the most populated value (according to the Boltzmann distribution).

To elaborate these, the following sentences are included in the revised manuscript:

Page 7, Line 10:

“In this work, because experiments showed that proton transfer in PBI membranes occurs effectively under acidic conditions² and theories suggest that fluctuation of the local dielectric environment can affect the proton transfer potential energy curves, $[\text{PBI}]_2$, $\text{H}^+[\text{PBI}]_2$ and $\text{H}_2^+[\text{PBI}]_2$ in $\epsilon = 1$ and $\epsilon = 23$ were chosen as model systems; the thermal energy agitations generally lead to fluctuations of the dipole orientation and local dielectric environment in the vicinity of molecules. The dielectric constants in the gas phase and bulk were used to simulate (approximate) low and high local dielectric environments, because the most populated value at/in the vicinity of the bifunctional H-bonds and the range over which it fluctuates at a particular temperature are not known.”

In this work, because we would like to systematically study the structures and energetics along the proton transfer curves in low and high local dielectric environments, the energy barriers (from electronic energies) and transition states were analyzed and discussed in detail first. Then, only the structures of the precursors (reactants) and transition states on the potential energy curves were used in TST calculations; TST calculations included the contributions of the thermal energies and partition functions etc. (e.g., eqn. (1) and (2)), from which the rate constants (k) and activation free energies (ΔG^\ddagger) were computed based on the Hessians of the precursors (reactants) and transition states. It is in our opinion not practical to calculate and discuss the free energies along the potential energy curves.

To give some information on the effect of the zero-point vibrational energy (ZPE), the energy barriers with the ZPE corrections are included in the revised manuscript (Table 1); the energy barriers with the ZPE corrections were used in the calculations of k^{Q-vib} , k^{S-Wig} and k^{F-Wig} . It appears that the trends of the energy barriers computed with and without the ZPE corrections are the same. We therefore include this finding in the revised manuscript.

Page 15, Line 10:

“It appears in Table 1 that the trends of the energy barriers without (ΔE^\ddagger) and with the zero-point vibrational energy corrections (ΔE_{ZPE}^\ddagger) are the same, for which ΔE_{ZPE}^\ddagger are systematically lower.”

To discuss the temperature dependence of the effect of local dielectric environment, the activation free energies (ΔG^\ddagger) in Table 1 are used and the following sentences are included in the revised manuscript.

Page 15, Line 12:

“Analysis of the free energy barriers (ΔG^\ddagger) in Table 1 supports the discussion made based on ΔE^\ddagger , for which the possibility for $[PBI]_2$ to be involved in the Grotthuss mechanism could be ruled out and formation of the transition structure ($CC-[2]^{2+,\ddagger}$) in the concerted bifunctional proton transfer in $H^{2+}[PBI]_2$ is thermodynamically less favorable in high local dielectric environment, e.g., at 333 K, $\Delta G^\ddagger = 38.2$ kJ/mol. Comparison of ΔG^\ddagger in $\epsilon = 1$ and 23 also confirms the trends of the effect of local dielectric environment, for which the increase/decrease in ΔG^\ddagger in $\epsilon = 23$ corresponds to the increase/decrease in the energy barriers (ΔE^\ddagger and ΔE_{ZPE}^\ddagger) at the transition states. For example, for $H^{2+}[PBI]_2$ in $\epsilon = 1$ and 23, $\Delta E^\ddagger = 6.3$ and 64.7 kJ/mol, and at 500 K, $\Delta G^\ddagger = 0.4$ and 36.1 kJ/mol, respectively.

The temperature dependence of ΔG^\ddagger appears to be the same for $[PBI]_2$ and $H^+[PBI]_2$, for which the increase in the temperature from 200 to 500 K leads to an increase in ΔG^\ddagger . In contrast, for $H^{2+}[PBI]_2$ in $\epsilon = 23$, the temperature increase leads to a decrease in ΔG^\ddagger from 39.4 to 36.1 kJ/mol. Analysis of the energetics in Tables S1[†] and S2[†] suggests that these findings could be associated with the net stabilization/destabilization effect of local dielectric environment on the bifunctional H-bonds, especially for the transition structures; while ΔE^{NSE} for $[PBI]_2$ and $H^+[PBI]_2$ are all positive, the values

for CC-[1]²⁺ and CC-[2]^{2+,‡} are negatives, $\Delta E^{\text{NSE}} = -44.2$ and -86.2 kJ/mol, respectively. The latter reflect a strong stabilization effect of local dielectric environment on the double protonated bifunctional H-bonds. The decrease in ΔG^\ddagger as the temperature increases could be attributed to the increase in the population of the shared-proton N...H⁺...N H-bond; our BOMD simulations on the imidazole system (H⁺[Im]_n, n = 2–4)¹² revealed that the intensity of the ¹H NMR chemical shift associated with the shared proton (24 ppm) increases as the temperature increases from 298 to 500 K and the oscillatory shuttling motion of the shared proton possesses lower vibrational frequency than the asymmetric N-H⁺...N H-bond.

Comment 3:

3) Probably because of my ignorance, it is rather obscure to me why the authors calculate the rate coefficients using the semiclassical limit (eq. 1) whose physical consistency is rather low (and I'm optimistic!!). In other words, what should be done in principle, it is to divide the internal degrees of freedom (dof) of the reactant and transition structure between semi-classical dof, e.g. internal rotations (see also the section concerning the phenyl ring rotations), and quantum dof. Subsequently, on this basis, one should calculate the partition function of the classical dof with a phase integral and using the standard formulations used by the authors in eq. 2 for the latter one. This is very complicated and eq. 2 is certainly adequate for the purposes of the authors. But what about eq. 1? Since some of the readers, like me, might have this doubt it would be good to insert a small clarifying sentence or a supporting note.

Response 3:

In this work, the rate constants were computed first at the semiclassical limits to study the applicability in the proton transfer process; both k^{Class} and $k^{\text{Q-vib}}$ were calculated based on the partition functions of the precursors (reactants) and transition states, eqn. (1) and (2) respectively. We have shown that k^{Class} are significantly different from $k^{\text{S-Wig}}$ and $k^{\text{F-Wig}}$, whereas $k^{\text{Q-vib}}$ are different only for the paths with high T_c, e.g., for the neutral dimer (Table 1). Therefore, we ruled out the possibility of using k^{Class} and $k^{\text{Q-vib}}$ and used only the Wigner corrected rate constants ($k^{\text{S-Wig}}$) in the discussions.

It should be noted that the quantum mechanical tunneling could be studied reasonably well using the quantum instanton method. However, for the present PBI system, preliminary reaction path optimizations using the quantum instanton method required extensive computational resources. Based on the observation that the crossover temperatures, T_c, (the temperature below which the transition state is dominated by quantum mechanical tunnelling) are all lower than the desirable operating temperatures of high-temperature fuel cells (~350 K), the approximation of the effect of the quantum mechanical tunnelling using the Wigner transmission coefficients eqn. (4) and (5) is in our opinion sufficient.

To elaborate these, the following sentences are added in the revised manuscript.

Page 8, Line 22:

" k^{Class} and $k^{\text{Q-vib}}$ were computed to study the possibility to use these semiclassical rate constants in the discussion of proton transfer in H-bond systems."

Page 9, Line 20:

"In addition, the rate constants with full Wigner correction ($k^{\text{F-Wig}}$)^{28, 29} were also computed to assess all of the rate constants calculated in this work; although the quantum effect (the proton tunneling), which generally leads to reduction of the energy barrier, could be studied more accurately using the quantum instanton method, our preliminary reaction path optimizations using this method required extensive computational resources and therefore not applicable for the present PBI system."

Page 16, Line 26:

"Because the quantum effect is not negligible and the trends for $k^{\text{S-Wig}}$ and $k^{\text{F-Wig}}$ are approximately the same above the operating temperatures of fuel cells (350 K or $1000/T = 2.86 \text{ K}^{-1}$), only $k^{\text{S-Wig}}$ will be used in further discussion."

Comment 4:

4) I hardly understand the meaning of the physical – not numerical!! – effect of the rate coefficients (a non-equilibrium property strongly related to relaxation phenomena) on the equilibrium constants (by definition independent from any relaxation or time dependent phenomena). I would expect, as already remarked in the previous point, that the equilibrium constant can be affected by the way in which the partition function (i.e. the free energies) are evaluated but, frankly, I need a further explanation for understanding what tunneling has to do with free energy. Rather, I would say that the good agreement observed by the authors in this respect is due to the good level of their calculations.

5) p. 14 "The results show in general that within the thermal energy fluctuation at room temperature ($RT = 2.77 \text{ kJ/mol}$).....". I agree with such a statement...but the authors are spanning temperatures rather different from the room temperature. Therefore this section is rather fuzzy and incoherent or, at least, incomplete

Response 4:

The authors agree with Reviewer 2 that in principle, the equilibrium constant and free energy (thermodynamics) are not directly related to the quantum mechanical tunnelling (kinetics). In this work, although the equilibrium constants were computed from the ratio of the rate constants in the forward and reverse directions, it was not our intention to discuss the correlation between the rate constants and free energies. The quantum mechanical tunneling usually increases the rate constants in both forward and reverse directions (through a decrease in the energy barrier). Therefore, the equilibrium constants, which were obtained in this work from the ratio of k_{forward} and k_{reverse} , are not affected by the quantum mechanical tunneling.

By the way, $RT = 2.77 \text{ kJ/mol}$ was actually at 333 K, not at the room temperature (298 K).

To correct this and include the information at 500 K, the following statements were used in the revised manuscript.

Page 17, Line 22:

“The results show in general that within the thermal energy fluctuation at 333 K ($RT = 2.77$ kJ/mol, near the operating temperature of high-temperature fuel cells, ~ 350 K), ω_1 and ω_2 can be distorted by at most approximately ± 14 degrees from the equilibrium structures (e.g., **CC-[1]²⁺**) and at 500 K ($RT = 4.16$ kJ/mol), ω_1 and ω_2 can be distorted by at most approximately ± 22 degrees. To study the effect of torsional motion, potential energy curves for bifunctional proton transfer in equilibrium structures with ω_2 rotated by ~ 14 degrees were constructed; only ω_2 was chosen because ω_1 and ω_2 in structure **CC-[3]²⁺** are equivalent (ω_1 and ω_2 in Fig. 8b are 36.9 degrees and 36.5 degrees, respectively).”

Reviewer: 2

Comments to the Author(s)

Comment 1:

This is nice study to study the mechanism of proton transfer in high-temperature polymer electrolyte membrane for fuel cells. They could determine the different pathways, stability of transition states and kinetics of reactions of bifunctional proton transfer in poly(benzimidazole) under their neutral and cationic forms.

Response 1:

In this study, we included the cationic forms of the PBI dimers, $H^+[PBI]_2$ and $H^{2+}[PBI]_2$. Because this study did not account for the role played by the phosphoric acid (H_3PO_4), which is an effective dopant in PBI membrane fuel cells, we are in the process to analyze the effects of the H-bond networks among PBI and H_3PO_4 in both neutral and acidic conditions.

Comment 2:

What are the real-life conditions where the values of the dielectric media can be 1 and 23? How is the conversion of the lowest and highest effects of the electric fields translated to dielectric constants of 1 and 23?

Response 2:

As discussed in Response 2 to the comment of Reviewer 1, we would like to study the role played by local dielectric environment surrounding the bifunctional H-bonds; how the energy barriers (potential energy curves), H-bond interaction energies and rate constants change with the local dielectric environment. Because we do not know the value of the local dielectric constant at/in the vicinity of the bifunctional H-bonds (which could be different from the bulk) and range over which it fluctuates at a particular temperature, we decided to study the effect using the lowest possible ($\epsilon = 1$ in the gas phase) and average ($\epsilon = 23$ in the bulk) values; the thermal energy agitations lead to fluctuations of

the local dielectric constant (due to fluctuations of the dipole orientations), and it is difficult to know the most populated value (according to the Boltzmann distribution). An experiment (<https://doi.org/10.1016/j.spjpm.2015.03.008>) suggested for example that, for cyclohexanone, ϵ could reduce from 15 to 12 over the temperature range of 293-343 K ($\Delta T = 50$ K).

Comment 3:

In the methodology: “for which the effect of the size was systematically investigated using the B3LYP/TZP method” it is not clear what is meant by the size. Also, what is the limit of the size where the double zeta vs the triple zeta basis sets are used?

Response 3:

The basis sets used in this work are “DZP” and “TZP”, which are the “double-zeta” and “triple-zeta” basis sets (two and three gaussian functions per atomic orbital, respectively); usually, the larger the basis set (the larger the number of the atomic orbital (AO) in the linear combination atomic orbital-molecular orbital (LCAO-MO) approximation), the better the results. In general, in our experience, it is not easy to find the limits of the size of the basis sets extension (the Hartree-Fock limit), especially for large systems. In this work, we just want to show that based on these two basis sets, the computed equilibrium structures are the same and if the relative potential energy curves and relative H-bond interaction energies are the objectives, both basis sets yield approximately the same results, especially when the counterpoise correction of the basis set superposition error (BSSE) is applied. These findings could be beneficial for theoretical studies on larger systems, for which computer resources are restricted.

To elaborate “the effect of size”, the following sentence is modified in the revised manuscript:

Page 6, Line 19:

“Because the PBI systems studied in this work are large, the B3LYP method was applied primarily with the DZP basis set (abbreviated B3LYP/DZP), for which the effect of the size of the basis set was systematically investigated using the B3LYP/TZP method and the counterpoise correction of BSSE.”

Comment 4:

Which double zeta and which triple zeta basis sets are used? Are they Pople, Dunning, etc.? Do they have diffuse functions, etc.?

Response 4:

For the TURBOMOLE software package, DZP and TZP are the basis sets of Prof. Ahlrichs (Univesitaet Karlsruhe) and coworkers. They are with polarization functions but without diffuse functions. The polarization functions are necessary to describe polarization of the electron density of the atoms in the proton transfer process.

H: DZP [2s1p|4s1p]; TZP [3s1p|5s1p]

C: DZP [4s2p1d|8s4p1d]; TZP [6s3p1d|10s6p1d]

N: DZP [4s2p1d|8s4p1d]; TZP [6s3p1d|10s6p1d]

As an example, B3LYP/DZP calculation on a neutral PBI dimer involves total number of primitive shells = 31; total number of contracted shells = 420; total number of cartesian basis functions = 908; total number of SCF-basis functions = 860, whereas the same calculation with TZP involves total number of primitive shells = 40; total number of contracted shells = 592; total number of cartesian basis functions = 1176; total number of SCF-basis functions = 1128. For a Linux cluster with 32 processors, a single-point DZP calculation on the same neutral PBI dimer requires cpu-time ~25 minutes and the total wall-time ~3 minutes, whereas the calculation with TZP in Linux cluster with 16 processors requires cpu-time ~1 hours 29 minutes and 53 seconds and total wall-time ~11 minutes and 27 seconds.

Because the PBI dimers considered in this work involve more than 76 atoms and geometry optimizations (potential energy curves/surfaces) and Hessian calculations had to be performed repeatedly, we have shown that the DZP basis set is the best compromise between the accuracy and computational resources. Additional diffuse functions on every atom could lead to enormous extra cpu-time even for single-point calculation.

To give the information on the basis sets used in this work, the following sentences are included in the revised manuscript:

Page 6, Line 19:

“Because the PBI systems studied in this work are large, the B3LYP method was applied primarily with the DZP basis set (abbreviated B3LYP/DZP), for which the effect of the size of the basis set was systematically investigated using the B3LYP/TZP method and the counterpoise correction of BSSE. The DZP (double zeta polarized) and TZP (triple zeta polarized) basis sets for the H atoms are [2s1p|4s1p] and [3s1p|5s1p] and those for the C and N atoms are [4s2p1d|8s4p1d] and [6s3p1d|10s6p1d], respectively; [A|B] denotes contracted (A) and primitive (B) Gaussian functions.”

Comment 5:

Why is B3LYP used as method? It does not account for dispersion which is important in the studied systems.

Response 5:

We are aware of the fact that PBI is a π system and the π - π interaction could be important, e.g., for the benzene dimer, the π - π interaction is responsible for example for the stacking and parallel displaced structures. However, for the PBI planar structures, the dipole moments of the monomer and dimers are relatively high, e.g., for the neutral monomer, dipole moment in $\epsilon = 1$, $\mu = 4.13$ D and in $\epsilon = 23$, $\mu = 5.87$ D, whereas the values for the single protonated monomer in $\epsilon = 1$ and 23, $\mu = 16.25$ and 19.84 D, respectively. Therefore, the dipole-dipole interaction dominates the π - π interaction in the PBI coplanar dimers.

To elaborate this the following sentences are included in the revised manuscript.

Page 6, Line 10:

“It should be noted that the PBI system is a π system, in which the π – π interaction could be important. However, because the dipole moments of the neutral and protonated monomers are high, the dipole-dipole interaction is mainly responsible for the bifunctional H-bond interaction in the coplanar dimers; for example, for the neutral monomer, the dipole moments (μ) in $\epsilon = 1$ and 23 are 4.1 and 5.9 D and those for the single protonated monomer are 16.3 and 19.8 D, respectively.”

Comment 6:

How can the results of this study be used practically? What is suggested to use as material for fuel cells? And are the suggested systems feasible to be implemented in existing fuel cells?

Response 6:

This research work is a theoretical study that focused on proton transfer mechanisms in bifunctional H-bonds and on some fundamental factors (the local dielectric environment, temperature and conditions (neutral and acid conditions etc.) that could affect the kinetics of the elementary processes. Because there are other factors not included in the model systems, the statement in the abstract and conclusion in the original manuscript:

“These theoretical results provide insights into the Grotthuss mechanism, which can be used as guidelines for future experiment and development”

is too optimistic. The authors would like to tone down the statement by using the following sentence:

Page 2, Line 24 (abstract) and Page 21, Line 2 (conclusion):

“These theoretical results provide insights into the Grotthuss mechanism, which can be used as guidelines for understanding the fundamentals of proton transfers in other bifunctional H-bond systems.”

Comment 7:

Legends in Figure 6 are too tiny to be seen, they need to be adjusted.

Response 7:

The fonts of the legends in Figure 6 is increased from 10 pt to 12 pt.

Comment 8:

“which can be used as a guideline for future experiments and the development of the same and similar systems” in the conclusion: very vague, needs more specifics.

Response 8:

As discussed in Response 6, this work is a theoretical study that investigated mechanisms and some fundamental factors that could affect proton transfer in bifunctional H-bonds. The findings are therefore basic knowledge that could be directly applied on other bifunctional H-bond systems.

The following sentence is used in the revised manuscript:

Page 2, Line 24 (Abstract) and Page 21, Line 2:

“These theoretical results provide insights into the Grotthuss mechanism, which can be used as guidelines for understanding the fundamentals of proton transfers in other bifunctional H-bond systems.”

Comment 9:

The conclusion is very long, it needs to be more concise.

Response 10:

To comply with Reviewer 2, the authors shorten the conclusion, from 699 to 640 words (8.4%).

The following texts and symbols in the original manuscript were deleted in the revised manuscript:

“Because both imide groups in PBI are preferentially protonated with considerably high protonation constants,”

“especially when the counterpoise correction for BSSE was included; therefore, discussions were based only on the B3LYP/DZP results”

“(ΔE^\ddagger)”

“($\Delta E^{\text{Rel,H-bond/CP}}$)”

“($\Delta E^{\text{Rel,Solv}}$)”

“Although the rate constants and conductivities reported in the literature vary over a wide range and relevant thermodynamic properties are restricted for PBI membranes,”